# First Eddy Covariance Flux Measurements of Semi Volatile Organic Compounds with the PTR3-TOF-MS

Lukas Fischer[1], Martin Breitenlechner[1,§], Eva Canaval[1], Wiebke Scholz[1,4], Marcus Striednig[2], Martin Graus[2], Thomas G. Karl[2], Tuukka Petäjä[3], Markku Kulmala[3], Armin Hansel[1]

[1] Institute for Ion Physics and Applied Physics, University of Innsbruck, Technikerstraße 25, 6020 Innsbruck, Austria

[2] ACINN, Institute of Atmospheric and Cryospheric Science, University of Innsbruck, Austria

[3] Institute for Atmospheric and Earth System Research / Physics, Faculty of Science, University of Helsinki, P.O. Box 64, 00014 Helsinki, Finland

[4] Ionicon Analytik Ges. m. b. H., Eduard-Bodem-Gasse 3, 6020 Innsbruck, Austria

§ Present address: University of Colorado Cooperative Institute for Research in Environmental Sciences (CIRES) at the NOAA Chemical Sciences Laboratory (CSL) Boulder, Colorado USA

*Correspondence to*: Lukas Fischer (lukas.fischer@student.uibk.ac.at), Armin Hansel (armin.hansel@uibk.ac.at)

**Abstract.** We present first eddy covariance flux measurements with the PTR3-TOF-MS, a novel proton-transfer-reaction mass-spectrometer. During three weeks in spring 2016 the instrument recorded 10 Hz data of biogenic volatile organic compounds above a boreal forest, on top of a measurement tower at the SMEAR II station in Hyytiälä, Finland. Flux and concentration data of isoprene, monoterpenes and sesquiterpenes were compared to the literature. Due to the improved instrument sensitivity and a customized "wall less" inlet design we could detect fluxes of semi-volatile and low volatile organic compounds with less than single digit picomol/m²/s values for the first time. These compounds include sesquiterpene oxidation products and diterpenes. Daytime diterpene fluxes were in the range of 0.05 to 0.15 picomol/m²/s, which amounts to about 0.25% to 0.5% of the daytime sesquiterpene flux above canopy.

## 1 Introduction

More than 1000 teragrams (Tg) organic carbon excluding methane is emitted from terrestrial ecosystems into the atmosphere per year as biogenic volatile organic compounds (BVOC) (Guenther et al., 2012). All anthropogenic VOC emissions together account for another 200 Tg $yr^{-1}$ of carbon flux into the atmosphere (Huang et al., 2017). Once in the atmosphere, these VOC react with the hydroxyl radical (OH), ozone ($O_3$), or nitrate ($NO_3$) radicals and subsequent $O_2$ addition generates peroxy ($RO_2$) radicals as intermediates. Further oxidative steps depending on the availability of NOx (NO + $NO_2$) lead to oxidized volatile organic compounds (OVOC) (Seinfeld and Pandis, 2016). These OVOC could degrade further through photooxidation reactions becoming less volatile and condense onto aerosol particles, contributing to secondary organic aerosol (SOA) or deposit to surfaces (Nguyen et al., 2015; Tröstl et al., 2016; Stolzenburg et al., 2018; Lehtipalo et al., 2018). Highly oxidized $RO_2$ resulting from monoterpene oxidation with subsequent H-shifts and $O_2$ addition were also found to produce accretion products having a remarkably low vapor pressure. These dimers are likely able to initiate new particle formation (NPF) (Ehn et al., 2012, Kulmala et al. 2013, Ehn et al. 2014, Kirkby et al., 2016, Tröstl et al. 2016, Berndt et al., 2018a, Berndt et al., 2018b). Resulting aerosol can act as cloud condensation nuclei and impact global climate as well as air quality (Pöschl, 2005). Molecular understanding of NPF and early growth is critical for climate modeling (Lee et al., 2019, Gordon et al., 2016). A

recent global model by Zhu et al., (2019) shows a decrease in total aerosol direct and indirect radiative forcing of 12.5% compared to previous models when including organic nucleation. Their calculations were based on an explicit chemical mechanism coupling emissions from the Community Earth System Model (CESM) with the IMPACT aerosol model. CESM itself uses global inventories of precursor emissions models like MEGAN (Guenther et al., 2012), extrapolated from local observations. These observations were gathered from numerous field-measurements that are limited to VOC fluxes, which are
detectable with current analytical devices. Proton transfer reaction mass spectrometry (PTR-MS) (Hansel et al., 1995) can detect the majority of volatile organic carbon in ambient air (Hunter et al., 2017). It is a well-established technique for direct emission measurements of VOC (Karl et al., 2001; Müller et al., 2010) and was used successfully in gradient and eddy covariance flux measurements (e.g., Millet et al., 2018; Karl et al., 2018; Rinne et al., 2007; Ruuskanen et al., 2011). VOC including methanol, isoprene and monoterpenes comprise the largest fraction of emissions and can be detected fast enough for
eddy covariance flux analysis by existing PTR-MS instruments. Still, Guenther et al. (2012) stresses the need for instrumentation able to measure BVOC with low vapor pressures and semi-volatile organic compounds (SVOC), in order to refine current emission models. This aim goes in line with monitoring highly oxidized organic molecules (HOM) important for NPF and SOA formation.

A group of compounds challenging current instrumentation are sesquiterpenes (SQT; $C_{15}H_{24}$), which can be highly reactive.
Despite their lower emission rates, the importance of sesquiterpenes for NPF was postulated long ago (Bonn and Moortgat, 2003), but emission estimates are still based mainly on enclosure measurements (Bourtsoukidis et al., 2018; Hakola et al., 2006; Helmig et al., 2007). At the boreal forest site at Hyytiälä, Finland, Rinne et al. (2007) predicted that sesquiterpene emissions are as high as 20% of monoterpene emissions at the leaf level (based on enclosure measurements) and that thereof 30-40% are chemical degraded on the way from emission to instrument inlets at tower height. These sesquiterpene fluxes were
below limit of detection (LoD) of the state-of-the-art instruments at that time when considering reactivity, adhesion to surfaces and inlet damping (Bourtsoukidis et al., 2018; Rinne et al., 2007). Recent particle phase measurements at a hemi boreal forest site in Estonia suggest that sesquiterpene oxidation products can substantially contribute to secondary organic aerosol mass (Barreira et al., 2021, under review).

Here, we present our technical approach and first results of eddy covariance flux measurements of VOC and SVOC with the
recently developed PTR3-TOF-MS (PTR3) (Breitenlechner et al., 2017). The instrument features a greatly improved sensitivity compared to standard PTR-TOF-MS devices, a prerequisite for measuring the lower ambient concentrations of SVOC. The strength of the PTR3 lies in detecting VOC, OVOC, and HOM with concentrations at sub pptv levels. The PTR3 bridges a gap between previous PTR instruments and the atmospheric pressure chemical ionization techniques such as Nitrate-CIMS or Iodide-CIMS (Riva et al., 2019): While PTR-MS could detect most precursor compounds and volatile oxidized
compounds, HOM could not be detected. For such compounds previous PTR instruments were either not sensitive enough or these compounds were lost in the inlet line. At the same time the highly sensitive atmospheric pressure chemical ionization methods with rather long reaction times compared to PTR3 need to be more selective, otherwise saturation due to primary ion depletion by highly abundant precursor compounds and secondary ionization reactions complicate quantification. The PTR3 was designed to fill this gap by shifting maximum measurable concentrations down to atmospheric relevant levels and thereby
lowering its limit of detection. Similar progress was made in the meantime with the benzene cluster cation chemical ionization mass spectrometer reported by Lavi et al. (2018).

## 2 Experimental

### 2.1 Site

Measurements were conducted at the SMEAR II station (Station for Measuring Ecosystem – Atmosphere Relations) in Hyytiälä, Finland (61°51'N, 24°17'E, 181 m above sea level; Hari and Kulmala, 2005; Kulmala et al., 2001) during April and May 2016. The surrounding managed stand is predominated by 50-year-old Scots Pines (*Pinus sylvestris*), with less than 1% other species like downy birch (*Betula pubescens*), grey alder (*Alnus incana*) and aspen (*Populus tremula*). Ground vegetation includes heather (*Calluna vulgaris*), lingonberry (*Vaccinium vitis-idaea*) and blueberry (*V. myrtillus*) and the dominating moss

species *Dicranum undulatum*. Similar boreal coniferous forests represent 8% of the Earth's surface (Hari and Kulmala, 2005; Kulmala et al., 2001). The PTR3 mass spectrometer was placed inside a measurement container on top of a tower, 35 m above ground, overtopping the canopy by about 15 m. Local winds were originating from all directions with no clear prevailing direction during the whole campaign as shown in the directional histogram in Figure 1. Wind speeds with an average magnitude of 2.5 m/s were observed. The inlet was oriented horizontally, pointing in southeast direction (165°). In close proximity of the

flux measurement tower (150 m away), various parameters such as temperature, humidity, photosynthetically active radiation, wind, trace gas concentrations, such as ozone, NOx etc. are routinely recorded at different heights of a second tower. These datasets are publicly available through the online Smart SMEAR service (Junninen et al., 2009).

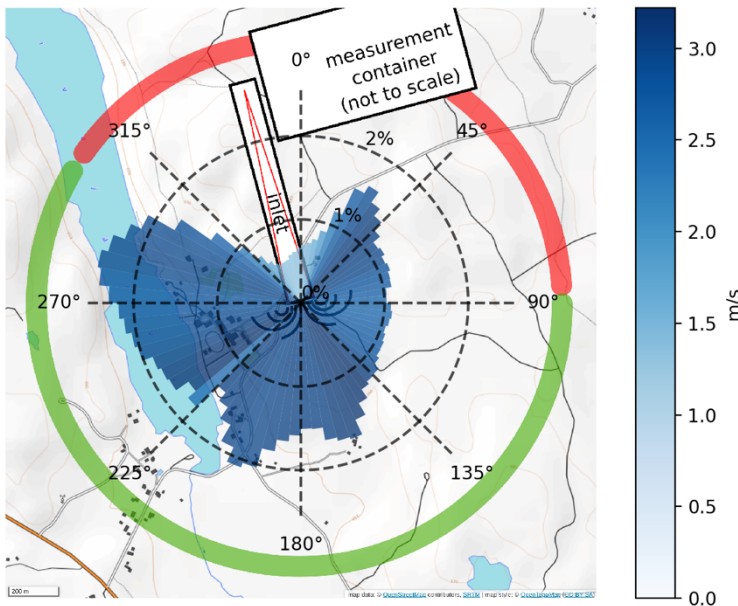

**Figure 1: Prevailing winds from April 26th to May16th in 2016. The orientation of the air inlet required filtering out data originating
from the wind sector shown in red: Length of the histogram bars equals occurrence frequency in the corresponding sector, average wind speed is indicated in blue. Data with wind directions within the green sector are considered undisturbed, the red sector is influenced by the container (indicated by the white box) on top of the tower structure. The orientation of the inlet structure described in detail later is also indicated by the long white rectangle. Wind data were recorded from a sonic anemometer located at the air intake point. Map: © OpenStreetMap contributors 2020, Distributed under a Creative Commons BY-SA License.**

## 2.2 Wind and humidity data

Horizontal and vertical wind components as well as virtual temperature were measured by a sonic anemometer (Metek USA-1) at 10Hz. The sensor was mounted 0.5 m above our air inlet, a trade-off between sensor separation and minimal disturbance by the high inlet flow. The sensor separation is still only 3.3% of the aerodynamic measurement height and according to Horst et al. (2009) we expect negligible flux attenuation in this configuration. An infrared gas analyzer (IRGA) from LI-COR Inc. (LI-840A) built into the mass spectrometer rack recorded water vapor concentration at a sampling frequency of 1Hz. The gas stream used for humidity measurements was extracted from the excess sample air of the core sampling flow with sampling lines shorter than 0.5m. We thereby ensured that the measurements represent the humidity of the same air stream entering the mass spectrometer. Synchronous recording of concentrations and wind data was assured by running the acquisition applications on the same computer and using the system clock for time stamps.

## 2.3 PTR3-TOF Mass Spectrometer (PTR3)

Highly time resolved VOC and SVOC volume mixing ratios were measured by the novel PTR3-TOF mass spectrometer described in detail by Breitenlechner et al. (2017). The PTR3 has a highly improved sensitivity and a special inlet design reducing wall losses for low volatility compounds compared to standard PTR-TOF instruments. In field calibrations were performed regularly by dynamic dilution of a gas standard (Apel Riemer) containing known amounts of different VOC in dry and humidified synthetic air. Impurities in the synthetic air were removed by a catalytic scrubber. We varied the water vapor concentration in the diluted calibration standard and in zero air by humidifying the synthetic air before entering the catalyst. In this way the calibration gas humidity was buffered and slowly changed from dry to humid conditions resulting in smooth ramps covering the whole humidity range. Absolute humidity was monitored simultaneously by the same infrared gas analyzer used for the ambient measurements. The measured dependence of the sensitivity for the calibration compounds on humidity was fitted by an empirical model, later used for calibration of the ambient measurements. The instrument operated with $H_3O^+(H_2O)_n$ (n=0-3) primary ions at a reduced electric field strength E/N of 80 Townsend and a pressure of 76 mbar in the reaction region. PTR3 sensitivities are highest for ketones and almost humidity independent as was shown by Breitenlechner et al. (2017). The ketones present in our calibration standard are therefore used as a measure for the maximum expected sensitivity for unknown compounds assuming fast reactions close to the collisional limit value with all $H_3O^+(H_2O)_n$ primary ions. To account for the mass dependent duty cycle of the TOF analyzer, we normalized measured in counts per seconds (cps) relative to the duty cycle at m/z 100, resulting in duty cycle corrected cps (dcps). The choice of this reference m/z where transmission is assumed to be 100% is arbitrary and does not influence calculated concentrations. Expected transmission of a compound of exact mass $M$ is then calculated by the square root of 100/$M$.

Hexanone and methyl ethyl ketone both showed a sensitivity of 16 ± 2 dcps/pptv (duty cycle corrected counts per second, per pptv) independent of humidity. With these settings, α-pinene could be ionized reasonably well with a sensitivity of 6.8±1.5 dcps/pptv at dry and 3.8±0.9 dcps/pptv at humid conditions (8 parts per thousand of $H_2O$). Isoprene sensitivities were 5.4±0.9 dcps/pptv at dry and 2.4±0.5 dcps/pptv at humid conditions. Fragmentation was kept low as indicated by α-pinene fragmentation, m/z 81 to m/z 137 was 32% on average. β-caryophyllene was not available as on–site gas standard, but it was calibrated in the laboratory at the same instrument settings and a sensitivity of 7±3 dcps/pptv was found independent of the humidity. The respective limits of detection (LoD) were 12 pptv for isoprene, 7.5 pptv for α-pinene and 1.7 pptv for β-caryophyllene at 1s integration time based on the 2σ standard deviation of the chemical background during calibration divided by the measured sensitivity. The measured LoD was mainly limited by the background in the available VOC free zero air. Mass spectra were recorded at 10 Hz up to m/z 610, mass scale calibration during data acquisition was done based on known

primary ions and a diffusion source constantly adding D5 siloxane ($Si_5O_5C_{10}H_{30}$) to the reaction chamber of the PTR3 producing an ion signal at the known protonated mass m/z 371.102.

## 2.4 Inlet design

Low volatility species are most affected by inlet line losses hence a more sophisticated inlet concept is required. Accurate eddy covariance flux measurements require air velocity, temperature, humidity and SVOC concentrations to be sampled co-located

and synchronized without disturbing the natural air flow (Aubinet et al., 2012, p.45ff., 72). Chemical ionization mass spectrometers are by design closed path analyzers and too bulky to be deployed within the preferable sensor separation distance of < 50 cm (Massman, 2000) to the sonic anemometer. Most previously reported tower flux measurement campaigns using mass spectrometers (Millet et al., 2018; Fulgham et al., 2019; Karl et al., 2018; Schallhart et al., 2017) extended the inlet lines from the top of the measurement tower down to the instruments placed in a container at ground level. In such setups, well

known limitations caused by long inlet lines (~50 m) of closed path analyzers have to be tackled. While high frequency attenuation of fluctuations created by these long inlet lines is sufficiently small for highly volatile compounds (Lenschow and Raupach, 1991), surface interacting molecules like SVOC are affected substantially (Massman, 1991). In order to avoid these drawbacks, we place the PTR3 on top of the measurement tower and deployed a new inlet concept for the remaining ~4 m. We sampled air as far away from the container as possible to reduce influences of the container structure on the wind flow

field while keeping wall contact of the sample air as low as possible. We have chosen the sampling line inner diameter (ID) and the air flow rate in such a way that the hydrodynamic entrance length is longer than the sampling line itself. That way we avoid that turbulence - developing at the shear regions close to the tubing wall - reaching the center of the tube where the core sampling is located as illustrated in Figure 2. The entrance length was calculated as $L_{h,turbulent} = 1.359D\mathrm{Re}^{1/4}$ as described by Zhi-qing (1982). Despite operating in highly turbulent conditions at Reynolds numbers (Re) above 42000, there is still an

undisturbed laminar cone stretching from the entrance of the tube down to the intake point of our core sampling line. By sampling from the center at the end of the tube, only the central portions of the flow, which have not contacted the inlet line wall, are analysed by the mass spectrometer. Our choice was a 20 cm ID pipe at a flow velocity of 3 m/s, resulting in a theoretical entrance length of 3.9 m. The sampling tube had to be mounted horizontally due to constraints imposed by other instruments. Anemometer deviation caused by the suction of the air intake are estimated to be below 2 cm/s at a sensor

separation distance of 0.5 m. This was estimated assuming a superimposed spherically symmetric flow created by the inlet. Since the anemometer is mounted above the inlet, this constant systematic flow is removed by the Reynolds decomposition of the vertical wind component during analysis.

The 20cm ID pipe consisted of double walled stainless steel with mineral wool insulation in between. While wall contact is already minimized by core sampling, the reflective metal surface and thermal insulation reduce heating of the exposed inlet

walls by sunlight and additionally avoid SVOC partitioning effects inside the inlet due to temperature changes. The sample flow of 3 m/s was generated by a blower mounted at the end of the tube 1 m downstream of the core sampling (see Figure 2).

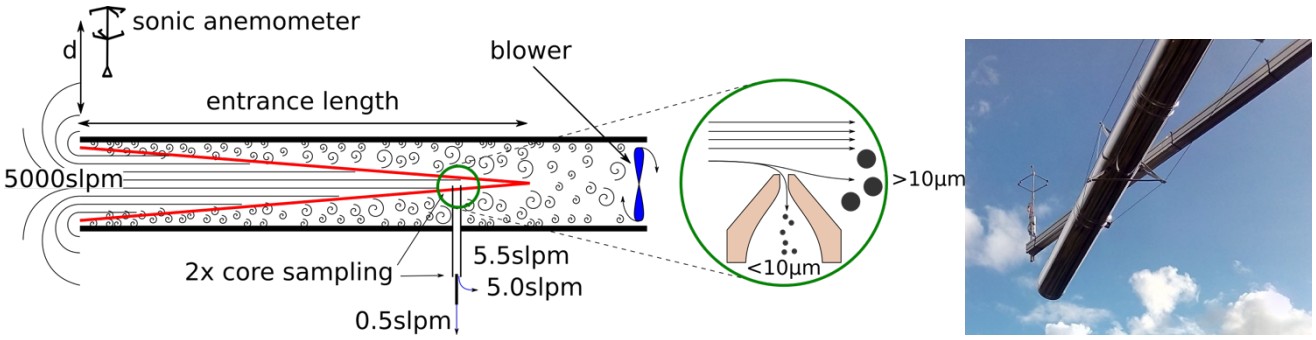

**Figure 2: Left: Inlet concept. A high flow of ambient air is sucked through a tube by a blower. The air inside the entrance cone (red) is mostly free of wall contacts and can be sampled from the center of the tube several meters away from the opening. Sampling perpendicular to the high inlet flow through a small nozzle creates a cut-off for particles >10µm. The sonic anemometer is placed at distance d=0.5 m above the air inlet. Right: Photograph of the inlet system.**

The remaining inlet of 0.4 m length from the center of the 20 cm ID tube through the container wall to the PTR3 consisted of a virtual particle impactor and a second center sampling setup. Through a critical orifice of 0.8 mm diameter, 5.5 slpm of ambient air was sampled perpendicular to the flow inside the large tube (see Figure 2, detail), resulting in an aerodynamic particle cutoff diameter of about 8 µm. This way we avoid clogging of the inlet and contamination of the ion source by larger particles like pollen and dust. Sampling efficiency for the second core sampling section was calculated to be larger than 93% for compounds which are completely lost at the wall surface according to Fu et al. (2019). For the calculation, we used the diffusion coefficient of α-pinene.

## 3 Data Processing

The analysis of raw data consisted of separate tasks described in the following section: reducing the raw TOF spectra to mass specific time traces, calculating concentrations from ion count rates and deriving eddy covariance flux from wind and concentration data.

### 3.1 Time trace calculation from mass spectra

Similar to Müller et al. (2010), data reduction of the raw time bin based TOF spectral data was performed in order to reduce cross talk of neighboring masses and speed up eddy covariance calculations. We used our Julia based analysis scripts (https://github.com/lukasfischer83/TOF-Tracer) already mentioned in previous work (Breitenlechner et al., 2017; Stolzenburg et al., 2018). Spectra were mass scale calibrated every 5 minutes and long term averaged for TOF peak shape analysis. For peak fitting and identification, we developed an interactive software with live visual feedback on the resulting fit quality, automatic isotope pattern calculation and molecular composition proposition and assignment (https://github.com/lukasfischer83/peakFit). This program was used on the averaged spectrum with the calculated peak shapes to create an optimized mass list of more than 1800 identified isobaric compounds with different sum formulas, excluding isotopes. Based on the resulting mass list, 10 Hz time traces were obtained by integration of non-overlapping intervals around each mass followed by deconvolution of interference caused by contributions of neighboring masses and isotopes in each spectrum. The script run time on an Intel Core i7 quad core processor system with SSD and 32GB of RAM is about 1 hour for 10 hours of 10 Hz data. The recorded amount of raw data in native HDF5 format was more than 30GB per day.

## 3.2 Humidity dependent calibration

It is extremely important to characterize the humidity dependent sensitivity of the PTR3 for different compounds. The new key features improving the PTR3-TOF sensitivity have been achieved with a 30 times longer reaction time and a 20 times higher pressure in the chemical ionization region. This means that the number of ion molecule collisions in the reaction zone has strongly increased compared to normal PTR-TOF-MS. Under these conditions, equilibria between forward and reverse reactions involving collisions between ions and water molecules have to be considered even at humidity levels in the single digit parts per thousand range (Breitenlechner et al., 2017). While $H_3O^+$ is the most prominent primary ion in standard PTR-TOF-MS (at 2 mbar and 80 Td) the primary ions in the PTR3 consist of a distribution of $H_3O^+(H_2O)_n$ cluster ions (n=0-3) that change as a function of humidity. In contrast to PTR-MS where proton transfer between $H_3O^+$ and VOC was the major ionization process, in the PTR3 ligand switching between $H_3O^+(H_2O)_n$ cluster ions and organic compounds becomes important. Many oxidized VOC (OVOC) react fast (i.e., reaction occurs with the collisional reaction rate) with all $H_3O^+(H_2O)_n$ ions (n=0-3), resulting in a rather small humidity dependence and a high sensitivity. This is particularly the case for compounds with high proton affinities and strong bond energies between the ligand and the (hydrated) hydronium ions like ketones as shown in Breitenlechner et al. (2017). In contrast, pure hydrocarbons react compound specific with individual $H_3O^+(H_2O)_n$ ions. Isoprene, for example, reacts with $H_3O^+(H_2O)_n$ ions n=0-1, while α-pinene reacts with ions n=0-2. This leads to a reduction of sensitivity and a moderate humidity dependence of typically a factor of two under ambient humidity. A factor of two in sensitivity change (less sensitivity at higher humidity) can cause an artificial deposition flux signal of such a compound even with an actually stable ambient concentration, modulated by water vapor emission fluxes. This potential error source in flux measurements was also discussed by Millet et al. (2018), who additionally compared errors resulting from different normalization methods to the primary ion signal. We chose to employ no normalization to the primary ion signal of the $H_3O^+(H_2O)_n$ ions and directly calibrate the individual product ion signals as a function of humidity.

Humidity dependent calibrations of the PTR3 for a set of individual compounds were done in the field every couple of days. A full calibration run needs about 1-2 hours, so a compromise between data loss and accuracy was made. We chose intervals based on observed instrument sensitivity stability and tried to avoid calibrations during periods with good flux conditions. Since the humidity of the ambient air sample changes at eddy frequency, calibrations had to be applied sample by sample at the recorded frequency of 10 Hz, before calculating the eddy covariance and performing an averaging. Each calibration measurement was parametrized as a function of sample humidity by fitting an empiric curve to calibration data (see Appendix A). For every 10 Hz ambient sample data point, a corresponding sensitivity was calculated and applied which requires fast humidity sampling. This fast humidity signal was reconstructed from a proxy ion signal reflecting the sample humidity. The relation between our proxy signal and the slower IRGA humidity time trace was recalibrated several times per day. A similar approach was demonstrated by Ammann et al. (2006), who used the $H_3O^+(H_2O)$ ion signal as proxy. In PTR3, the signal of $N_2H^+$ showed a much better correlation to the humidity measurement of the IRGA. $N_2H^+$ is created in the PTR3 by endothermic proton transfer from $H_3O^+$ ions in collisions with $N_2$ at low pressure in the transition region to the TOF quadrupole interface. Exothermic proton transfer from $N_2H^+$ to $H_2O$ in the sample gas is the predominating loss reaction. Consequently, the $N_2H^+$ signal depends inversely on the absolute humidity. Results are shown in figure A2.

## 3.3 Eddy covariance flux calculations

Eddy covariance flux calculations are based on the innFLUX code (Striednig et al., 2020). The calibrated concentration data were exported together with wind and IRGA data to conform to the necessary input format of innFLUX. Three similar data sets were exported for the different sensitivity estimates mentioned earlier and analyzed separately in order to study the effects of calibration uncertainties. The analysis routines of the innFLUX code include sector dependent tilt correction for wind data, lag time determination and calculation of several quality test parameters that are described in detail by Striednig et al. (2020).

An ensemble average of 30 min was selected as a compromise between data loss due to non-stationary conditions and artificial low frequency attenuation (Lee et al., 2005, p.20). No spectral corrections were applied, but cospectra are compared to the sonic anemometer sensible heat flux in the discussion section. Since the PTR3 was calibrated adding humidity to a dry, known standard gas concentration, the humidity dependent calibration already includes the additional dilution by the water vapor. Therefore, no WPL correction is necessary.

## 3.4 Spectral analysis

Inlet performance can be assessed by comparing the high frequency attenuation in the spectrum of the eddy covariance flux contributions to those of sensible heat (Aubinet et al., 2012, p.93ff.). While Millet et al. (2018) was using an improved PTR-QiTOF-MS having sensitivities closer to the PTR3, both, Park et al. (2013) and Millet present sesquiterpene fluxes that can be compared to our results. Millet estimates <25% high frequency damping for sesquiterpenes and monoterpene oxidation products based on cospectral similarity with sensible heat flux w'T'. Our system allows to measure sesquiterpene ($C_{15}H_{24}$, protonated exact mass 205.195 Da), a sesquiterpene oxidation product ($C_{15}H_{24}O_3$; exact mass 253.180 Da) and probably the monoterpene oxidation product pinonaldehyde ($C_{10}H_{16}O_3$; exact mass 169.122 Da) virtually as good as the volatile precursors like isoprene ($C_5H_8$; protonated exact mass 69.070 Da) as demonstrated in the ogive analysis in figure 3. Ogives represent the relative cumulative contribution of all measured eddies up to a certain frequency. At the frequency range carrying most of the eddy covariance flux, our setup shows no visible damping compared to w'T' which was directly measured with the sonic anemometer. This indicates the effectiveness of our wall contact reduced inlet concept and the overall fast time response of our setup and the PTR3. Ogives for the compounds measured by the PTR3 were calculated by cumulative numerical integration of the scaled, averaged cospectra passing the basic quality tests mentioned earlier and exceeding a signal to noise ratio of three. The ogive of w'T' was normalized to show a maximum of 100%. The other ogives were scaled to match at the frequency of the maximum covariance of w'T' indicated by the grey bar. Any substantial high frequency dampening would result in ogives flattening out below 100%. Shown in the plot are the filtered, averaged cospectra used for the ogive integration representing data from April 25th to May 17th.

Power spectra of the time traces are presented in Appendix B. They show contributions of white noise dominating the frequency range above 0.1 Hz which is typical for instruments based on count rates.

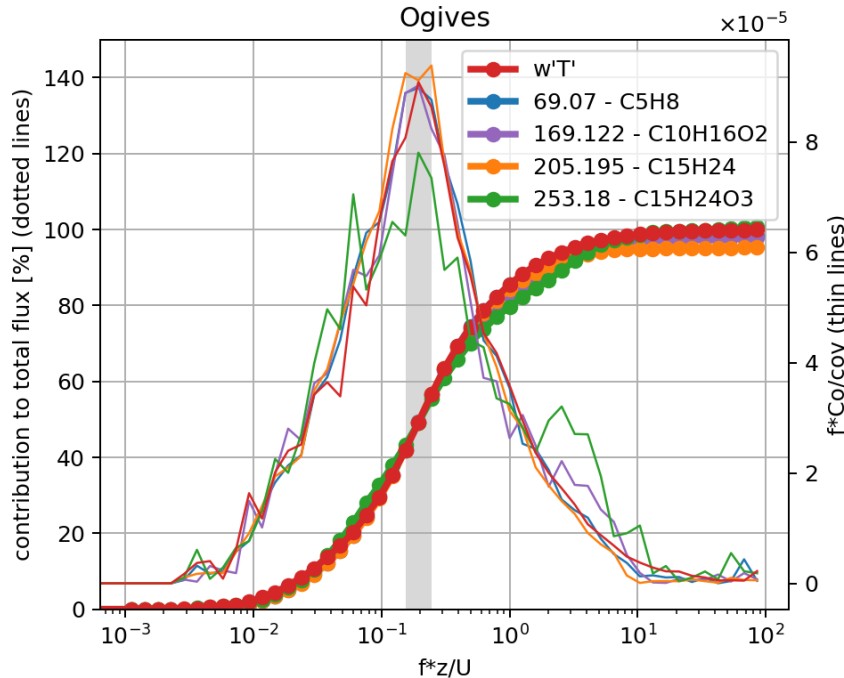

**Figure 3: Ogives (dotted lines) showing the cumulative flux contribution by frequency in percent, covariances *f\*Co/cov* (thin lines)**
**scaled to show spectral contributions. The grey bar indicates where the ogives were scaled to intersect with the ogive of w'T'. Red**
**trace shows sensible heat flux w'T' as best reference, no high frequency dampening is visible for all compounds including also semi**
**volatile $C_{15}H_{24}O_3$.**

## 3.5 Lag times

For the low concentrations that are typical for sesquiterpenes and sesquiterpene oxidation products, lag times between the
sonic anemometer data and the mass spectrometer data become harder to determine. Striednig et al. (2020) proposed to ensure
consistent lag times by experimental design, so covariances of individual ensemble average intervals can be accumulated for
a sharper maximum. Their routines in innFLUX already provide lag time determination from accumulated covariances, which
were used in our analysis. Figure 4 shows an example, which was obtained averaging over a period of eight hours on May 13[th].
The presented compounds differ greatly in volatility, and their very similar lag times indicate an efficient wall contact reduction
of our inlet design. Comparing to the covariance shape of sensible heat measured directly by the sonic anemometer further
indicates that little attenuation is created. The lag time of isoprene throughout the campaign during times with valid flux
conditions was $2.34 \pm 0.36$ seconds.

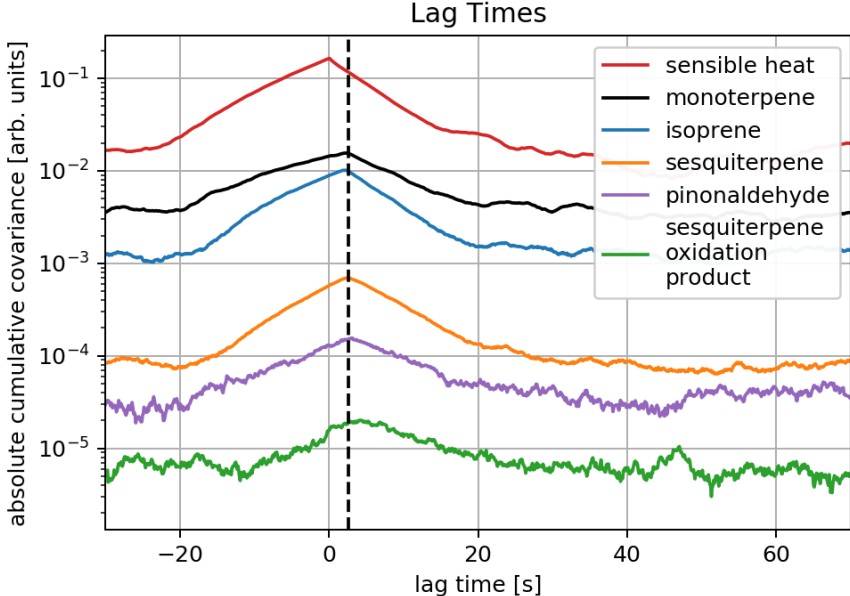

**Figure 4: Lag times calculated from absolute covariances averaged over a period of 8 hours on May 13th. The black vertical line illustrates the very similar lag time of all compounds having different volatility.**

### 3.6 Uncertainties

The compounds presented here have been chosen to cover a wide range in concentration, volatility and flux. Different sources of uncertainty dominate the individual compounds. While monoterpene and isoprene show decent signal-to-noise ratios in concentration data, their main source of uncertainty is the instrument sensitivity on the various detected tracers. For compounds present in the calibration gas standard, the corresponding calibration was applied directly. A total calibration error of ±30% including uncertainties of the gas standard mixing ratio, dilution errors and sensitivity drifts of the instrument between calibration intervals was estimated. For the other compounds a best estimate, an upper and a lower limit for sensitivity were assigned. The upper limit corresponds to an assumed ionization at the collisional limit in the reaction region. It was derived from the calibrations of 2-hexanone with a safety margin of +30%, the lower limit was chosen depending on the sum formula of the uncalibrated compound. We assumed that at least 50% of the 2-hexanone sensitivity for compounds containing more than ten carbon atoms and one or more oxygen atoms, as well as for compounds with at least five carbon atoms and more than two oxygen atoms. Compounds with a) at least two oxygens or b) at least five carbon atoms and one oxygen atom or c) more than ten carbon atoms and no functional group were treated similar to α-pinene. All remaining compounds were calibrated with the sensitivity of acetonitrile, which shows the least favorable water dependence. However, in this publication, we only discuss two compounds quantitatively using estimated sensitivities: $C_{15}H_{24}O_3$ and $C_{20}H_{32}$.

Therefore, our data usually represent the lower limit of concentrations or emissions, with a positive estimated uncertainty based on composition. Artificial flux contributions from water vapor emission due to humidity dependent ionization efficiency would cause a deposition flux bias. This means that for uncalibrated compounds, an observed emission flux can be underestimated at worst, while observed small deposition fluxes could in principle be caused entirely by a humidity dependent

sensitivity. Exact humidity dependent calibration is most crucial for compounds with a slow emission or deposition velocity, since their relative concentration fluctuations due to flux are small and can be masked by sensitivity modulation caused by water vapor flux. An example for this effect is given in Appendix A, where we compare fluxes derived from the same raw scalar traces using different humidity signals for calibration.

Random flux error estimates given in the discussion of the individual compounds were computed by innFLUX, which implements several different estimates proposed in the literature. We chose the method by Finkelstein and Sims (2001), which seemed to give the most conservative error estimate on our dataset. This method is based on calculating the "variance of a covariance, which includes auto- and cross-covariance terms for atmospheric fluxes", in the case of an already known lag time. Although, only for the very low fluxes of diterpenes, the random flux noise contributes significantly compared to the instrument sensitivity uncertainty.

We did not consider additional errors attributed to uncertainties in lag time determination, since our inlet concept theoretically does not introduce a compound, temperature or humidity dependent retardation due to its contact reduced design. Lag times determined quite precisely from a compound with good signal to noise ratio can therefore be used as a prescribed lag time for noisier signals.

## 4 Results and Discussion

### 4.1 Campaign overview

Flux data acquisition was started on April 17[th], but rain and snowfall prevented good flux conditions during the first week. In the time period from April 25[th] to May 16[th] 2016 we were able to record flux and volume mixing ratio data during 97% of the time. Exemplary data are shown in Figure 5, accompanied by the SMEAR II station data for ozone ($O_3$), nitrogen oxides (NOx), temperature (T) and photosynthetically active radiation (PAR) at 36 m. We filtered these data for a minimal friction velocity (u* > 0.3 m/s) to assure turbulent conditions discarding about 30% of potential flux data, mostly during nighttime hours. Additionally, we tested stationarity for each compound individually as proposed by Foken and Wichura (1996). Measurements with wind coming from the sector between 300° and 90° were rejected, because they might have been influenced by the tower structure (see Figure 1). This sector was identified based on wind inclination deviation from the horizontal plane shown by the results of the tilt correction calculation. After filtering, 60-70% of recorded data remain as indicated by the dark-colored periods of the overall time traces shown in Figure 5. Validity of data based on u* and wind direction is indicated on top of the flux data plot by the red and black bars, respectively. Recorded ion signals were assigned to most likely compounds according to the identified sum formula of protonated species.

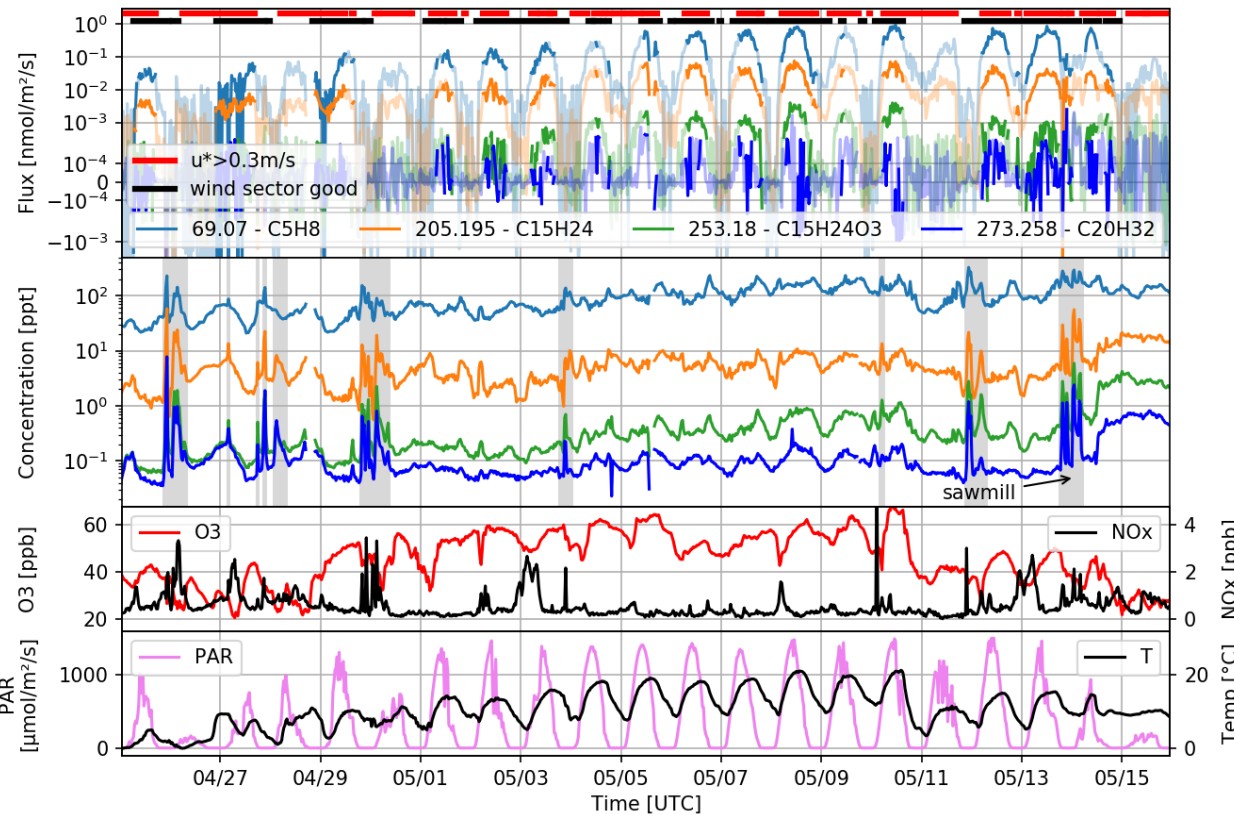

**Figure 5: Campaign overview.** Eddy covariance flux and concentration data of isoprene ($C_5H_8$), sum of sesquiterpenes ($C_{15}H_{24}$), sesquiterpene oxidation products ($C_{15}H_{24}O_3$) and sum of diterpenes ($C_{20}H_{32}$) Data filtering based on friction velocity, direction and stationarity is indicated by light colors of the flux traces, filtering on friction velocity and wind sector are indicated independently on top by red and black bars. Periods with concentrations influenced by local sawmill emissions are marked in grey. $O_3$, NOx and temperature recorded at 36m height as well as light are shown in the two bottom panels.

We observed periods showing highly elevated concentrations of isoprene, the sum of sesquiterpenes and the ozonolysis products of sesquiterpenes and diterpenes ($C_{20}H_{32}$; protonated exact mass 273.258 Da) together with numerous other molecules (not shown in Figure 5). Wind back trajectories during these events calculated from our sonic anemometer readings enabled us to pinpoint the emission source to the nearby sawmill facilities in Juupajoki located 6,4 km southeast of the Hyytiälä flux tower. Such events are indicated by grey areas in the concentration time trace in Figure 5. Eerdekens et al. (2009) also observed increased concentrations of certain compounds measured in Hyytiälä originating from the sawmill. Flux footprint contributions were calculated with the method published in Kljun et al. (2015). We see that the forestry field station 400m west-southwest of measurement tower can contribute to our flux measurements at corresponding wind directions. Depending on wind conditions, the station lies within an area which contributes 60% to 80% to the total measured flux.

About 200 further compounds excluding isotopes show covariance with the vertical wind component in our dataset. While two previous publications report even higher numbers (Millet et al., 2018; Park et al., 2013), ambient temperatures during their measurement periods varied between 15°C (Night) and 40°C (Day) and were substantially higher compared to our site and season (0°C-20°C). A study that may serve as a better comparison was performed in Hyytiälä also in May, but in 2013, published by Schallhart et al. (2017). They reported only 12 compounds above flux detection limit using the same generation of PTR-TOF as Park et al., 2013.

## 4.2 Isoprene

Isoprene is a VOC with well characterized sensitivity due to frequent calibrations during the campaign. We provide isoprene as a reference for comparison to more challenging compounds. Its atmospheric lifetime is long enough that chemical degradation during turbulent vertical transport from the source to our measurement tower height is negligible (Rinne et al., 2007). Our measured isoprene concentrations exhibit a more complex diurnal pattern, that is influenced by emissions, chemistry and boundary layer growth. In contrast, isoprene fluxes increase during the day following light and temperature peaking at around noon. Isoprene emissions decrease by one and a half orders of magnitude during sunset before eddy covariant flux conditions are no longer met due to low turbulence. Daily flux maxima are increasing from $0.045\pm0.025$ nmol/m$^2$/s in late April to $0.94\pm0.44$ nmol/m$^2$/s in May. This rise, which can be attributed to the onset of the growing season, nicely follows the increasing maximum temperatures during the first weeks in May. Measurement error contributions are dominated by the assumed calibration uncertainties ($\pm0.41$ nmol/m$^2$/s), while flux calculation error based on the method of Finkelstein and Sims (2001) is smaller ($\pm0.14$ nmol/m$^2$/s).

Schallhart et al. (2017) report an average isoprene flux of 0.035 nmol/m$^2$/s in 2013. We observed an average flux of $0.146\pm0.089$ nmol/m$^2$/s during the first half of May in 2016. The average temperature during our measurements in the first two weeks of May was more than three degrees higher than the average of May 2013. Average photosynthetically active radiation was 443 μmol/m$^2$/s which is almost 50% higher compared to Schallhart et al. (2017) as well. According to the parametrization of Guenther (1997) this explains a factor of two higher isoprene emissions during our measurements. The remaining discrepancy is attributed to interannual variability including snow cover and foliar density of scattered deciduous species as well as instrumental uncertainty. Increased emission of methylbutenol (MBO) by Scots pine was reported by Aalto et al. (2014) during bud and shoot growth in springtime. Rantala et al. (2015) claim that isoprene is the dominant emission measured on m/z 69.070 in late summer, produced by spruce, aspen and willow within the footprint. Due to fragmentation, MBO is detected at the same exact mass as isoprene. Therefore, we cannot exclude contributions of MBO to the detected signal at m/z 69.070, which we termed isoprene.

## 4.3 Monoterpenes

Previously reported monoterpene ($C_{10}H_{16}$; protonated exact mass 137.133 Da) emission fluxes by Hakola et al. (2006), Rinne et al. (2007) and Schallhart et al. (2017) obtained at the same measurement site are only partially reproduced by our data. Our monoterpene flux data show sudden changes from emission to deposition most of the time, even when comparing two consecutive flux values. Each flux value is an ensemble average of a 30 min interval and is validity checked concerning u* and wind direction. Only for monoterpenes we observe a strong directional concentration gradient during many averaging intervals. A comparison of monoterpenes and isoprene is shown in figure 6 panel a) and b). The plots represent the 30 min time interval highlighted in red in panel c). The length of the bars in the wind rose plot show the percentage of time the wind is coming from the respective sector. Average concentrations within the sectors are encoded in blue. Such gradients in the monoterpene data suggests that emissions/depositions show strong inhomogeneities within the footprint at this time of the year in 2016. These unknown inhomogeneities cause monoterpene data to show high correlation with horizontal wind components at lower frequencies similar to data reported earlier by Yang et al. (2013) for OVOC measured at a coastal site. The steady state test fails for most averaging intervals reducing our data coverage for monoterpenes to less than 32%. Figure 6 panel c) shows validity checked monoterpene flux data colored in solid blue, while rejected data are shown in light blue. Errors calculated analogous to the isoprene data are represented by the blue shades. Even during periods which passed the steady

state test, the discussed inhomogeneities exist. The time trace in panel c) shows a time period on May 8$^{th}$, capturing a deposition event. Monoterpene data are still passing the steady state test, with the directional histogram in panel b) clearly including the aforementioned monoterpene source. Unfortunately, the prevailing wind direction included similar sources from several directions on most days of the campaign. The direction of the highest monoterpene concentrations seen in figure 6, panel b) and our calculated footprint extent (not shown) suggest that the forestry field station could be a local source of monoterpene flux. Further analysis is needed to characterize the composition of emissions from the station, which is likely coupled to daily routines such as cooking or woodworking.

Nevertheless, there remains a period from 05/12 noon to 05/13 noon with almost continuous valid data. During these two days, the wind direction was stable between 150° and 200° and no directional gradients in monoterpene concentrations were observed. Ambient temperature had dropped by 5 degrees compared to previous days. We measured a 24h average monoterpene flux of $0.29\pm0.19$ nmol/m$^2$/s, which is in agreement with 0.38 nmol/m$^2$/s reported by Schallhart et al. (2017) during the start of the growing season. Again, random flux error is a minor contributor to the given uncertainty with 0.01nmol/m$^2$/s. Taipale et al. (2011) also report median 24h average monoterpene emissions measured during May 2007 of $0.28\pm0.05$ nmol/m$^2$/s. While isoprene emissions were affected by a seasonally different start of the growing season during the compared observation periods, monoterpene seems less influenced. This is attributed to the lower temperature during the two days used for monoterpene analysis. Rinne et al. (2007) provides monoterpene flux data, although measured later during summer. Using their published $E_{30}$ normalized emission potential of 1.55 mg/m$^2$/h and temperature dependence of 0.11 C$^{-1}$ on the temperature measured in our study during the afternoon hours of 5/13, we obtain an emission flux of 0.62 nmol/m$^2$/s at 15.2°C, which is comparable to our measured flux of $0.74\pm0.49$ nmol/m$^2$/s including a random flux error of 0.05 nmol/m$^2$/s.

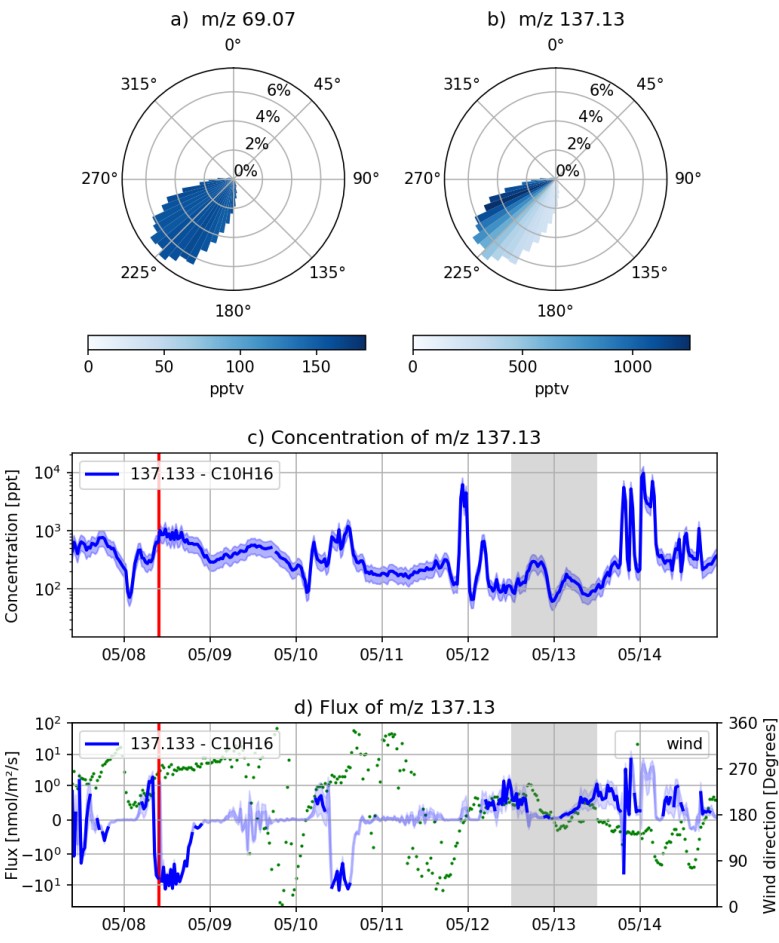

410

**Figure 6: Isoprene (C₅H₈, panel a) and monoterpene (C₁₀H₁₆, panel b) concentrations in parts per trillion (pptv; blue color coded) as a function of occurrence frequency (represented by sector length). Monoterpene concentrations are plotted in panel c) with uncertainties indicated by the light blue shading. The monoterpene concentration (panel c) and flux signal (panel d) are strongly fluctuating with wind direction. The red line indicates the 30 min interval used for averaging the 10 Hz wind and concentration data for the directional histograms a) and b). Only the grey period is used for comparison with literature monoterpene flux values. In the flux plot, periods indicated by a light blue line do not pass our quality checks for wind direction and friction velocity or the steady state test.**

### 4.4 Sesquiterpenes

Hakola et al. (2006) and Rinne et al. (2007) estimated sesquiterpene emissions to be 20% of that of monoterpene emissions based on enclosure measurements, with monoterpenes peaking earlier in June and sesquiterpenes dominated by β-caryophyllene in July. Due to the fast ozonolysis rate of β-caryophyllene, stochastic Lagrangian transport model calculations by Rinne et al. (2007) predict that only 30-40% of β-caryophyllene leaf level emissions reach the intake point on top of the measurement tower. Direct eddy covariance flux measurements with previous PTR-MS instruments did not show flux values even during highest emissions in summer due to instrumental constraints (LoD and sensitivity). Due to the high sensitivity of the PTR3 instrument we are able to report first eddy covariance fluxes of sesquiterpenes and sesquiterpene oxidation products at this site. Figure 7 shows diurnal sesquiterpene concentrations and fluxes. Concentrations are in the range of 1 to 10 pptv

slightly increasing during the campaign, matching data presented by Hellén et al. (2018). Fluxes show a pronounced diurnal pattern peaking in the early afternoon (local time) with maximum values rising from 4[+7,-3] pmol/m$^2$/s at the end of April to 54[+78,-40] pmol/m$^2$/s in mid of May. The eddy covariance flux analysis errors based on the method of Finkelstein and Sims (2001) are ±0.9 pmol/m$^2$/s for the lower and ±4.9 pmol/m$^2$/s for the higher flux value. Again, the majority of the measurement error is originating from the sensitivity uncertainty. Since the isomeric composition contributing to the PTR3 sesquiterpene ion signal is not known, we rely on calibrations obtained for β-caryophyllene after the campaign in the laboratory with the same PTR3 settings.

The chemical structures of the observed sesquiterpenes cannot be identified by PTR-MS. Previous enclosure measurements in Hyytiälä (Hakola et al., 2006) and in situ gas chromatography measurements (Hellén et al., 2018) reported, that β-caryophyllene is the most abundant sesquiterpene species emitted by Scots Pine. Hellén et al. (2018) claim that due to the short atmospheric lifetime of β-caryophyllene, O$_3$ reactivity and secondary organic aerosol production rates are dominated by sesquiterpenes in Hyytiälä during summer. In hemi boreal forests similar findings are reported from recent particle phase measurements (Barreira et al., 2021, under review).

Our eddy covariance flux measurements give additional insight in the local emission rates and chemistry of sesquiterpenes. These measurements show a substantial contribution of C$_{15}$H$_{24}$O$_3$ which was attributed to sesquiterpene ozonolysis products. From previous publications about the measurement site, β-Caryophyllene is known to have one of the highest sesquiterpene emission rates (Hakola et al., 2006; Hellén et al., 2018; Rinne et al., 2007) and is also very reactive (Richters et al., 2015), so it is presumably the predominant source of C$_{15}$H$_{24}$O$_3$. We suspect this signal to represent several isomeric first-generation ozonolysis products of β-caryophyllene based on published laboratory measurement results by Winterhalter et al. (2009). According to their findings, the products still contain an exocyclic double bond, resulting in atmospheric lifetimes shorter than those of for example α-pinene. Starting with a 1.5 times longer carbon backbone and a higher oxidation state compared to monoterpene, their oxidation products contribute substantially to aerosol particle growth (Li et al., 2011).

Here we estimate the fraction of sesquiterpene that was chemically converted during transport based on direct measurements of the products. The yield of C$_{15}$H$_{24}$O$_3$ from β-caryophyllene ozonolysis for the calculation was experimentally determined in a flow reactor experiment in our laboratory, using the same instrument at similar settings and conditions. Details on the measurement can be found in Appendix C. The yield of C$_{15}$H$_{24}$O$_3$ plus its most abundant fragments found after a reaction time of 12 seconds was 43±10 percent of the reacted β-caryophyllene. In the flow reactor experiments a similar fragmentation pattern of the ozonolysis products as in the ambient air measurements was observed. This can be interpreted as further evidence that β-caryophyllene is the precursor of our detected C$_{15}$H$_{24}$O$_3$ signal in ambient air. The reacted amount of β-caryophyllene is estimated from the product C$_{15}$H$_{24}$O$_3$ and its fragments flux divided by their yield presented in figure 7 (top panel). The reacted fraction (middle panel) is calculated dividing by the sum of measured tower level sesquiterpene plus the calculated reacted β-caryophyllene. The fraction of sesquiterpene that was converted seems mostly related to ambient temperature, while transport time which is related inversely to u* seems less influential. This could mean that most of the β-caryophyllene is converted in canopy and not limited by the transport time from the top of the canopy to our inlet on the measurement tower and that the ratio is mostly dependent on a temperature dependent sesquiterpene source composition. Another interpretation could be that temperature dependent in-canopy sinks determine how much C$_{15}$H$_{24}$O$_3$ leaves the canopy. Although further investigation seems necessary to fully explain our observations, these calculations are presented to give a perspective on what kind of topics can be researched with our setup. The ability to follow oxidation processes by monitoring precursor as well as product fluxes directly with one single instrument at these low emission rates is unprecedented and could help to tackle questions on sesquiterpene in- and above-canopy chemistry. Jardine et al. (2011) and Bourtsoukidis et al. (2018) for example emphasize the influence of sesquiterpenes on the oxidative capacity of the atmosphere but also struggle with technical limitations measuring sesquiterpenes and rely on enclosure measurements for emission estimates.

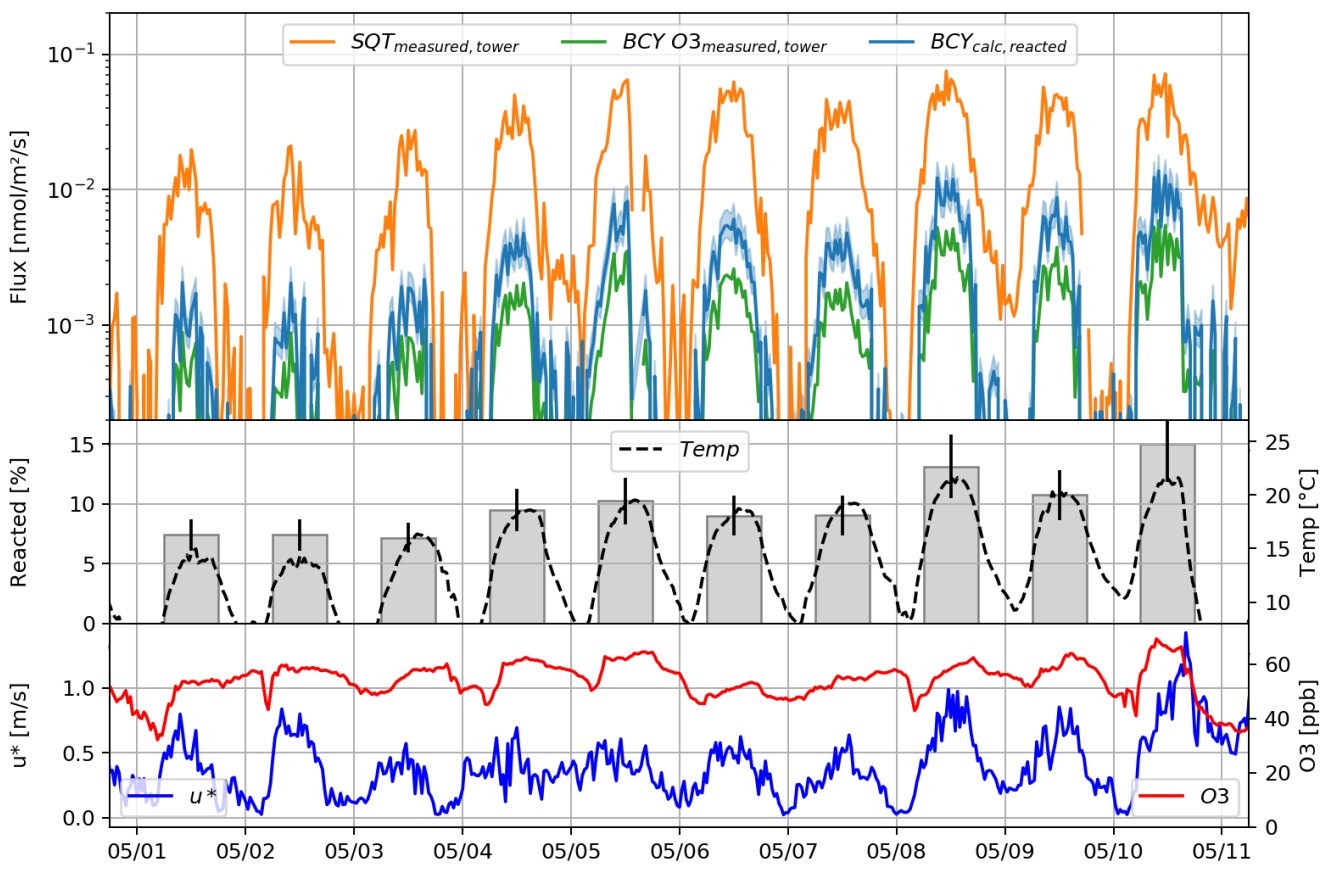

**Figure 7: Upper panel:** Calculated β-caryophyllene reacted (BCY$_{calc,reacted}$, **blue**) derived from measured oxidation products (BCYO3$_{measured,tower}$, **green**) compared to tower level sesquiterpene (SQT$_{measured,tower}$, **orange**). **Middle panel:** Daytime averages (8:00 am and 6:00 pm ) of the reacted fraction of sesquiterpene during transport (grey). **Bottom panel:** O$_3$ concentration and friction velocity u\*. The errors in blue shading for BCY$_{calc,reacted}$ and black errorbars for the reacted fraction indicate the systematic uncertainty originating in the yield used for the calculation.

### 4.5 Diterpenes

To the best of our knowledge, we can report the first direct eddy covariance flux measurements of diterpenes using mass spectrometry. We measured diterpene concentrations on $C_{20}H_{32} \cdot H^+$ in the order of 100 ppqv, which was below the limit of detection of previous PTR mass spectrometers. Complementary technologies capable of measuring such low VOC concentrations like Acetate-, Iodide- or Nitrate-CIMS are lacking ionization efficiency for pure hydrocarbon compounds (Riva et al., 2019). The recently developed benzene cluster cation chemical ionization mass spectrometer (Lavi et al., 2018) uses a similarly effective ionization method, but no data on limit of detection or field application data has been reported yet.

We aggregated  our flux measurements from April 28th to May 16th and calculated an average diurnal pattern of diterpene emissions shown in figure 8. Before this date, emissions were undetectable, probably due to low temperature. Similar to sesquiterpenes, the diterpene flux shows its peak in the early afternoon at an average of 0.15 pmol/m²/s with vanishing

emissions during night time. Since no calibration data or proton affinity data was available for the unidentified diterpenes, we assume ionization at the kinetic limit and thus give a lower estimate of the diterpene emission flux. This seems justified, if we extrapolate the increasing trend in measured ionization efficiency from isoprene via α-pinene to β-caryophyllene, probably due to the longer carbon backbone and thus higher proton affinity. The error bars do not include statistical error from day-to-day variations yet, those are shown in the comparison in Appendix B. Due to the short measurement period that we average, these variations are mostly responsible for the observed noise. No filtering for flux conditions was applied.


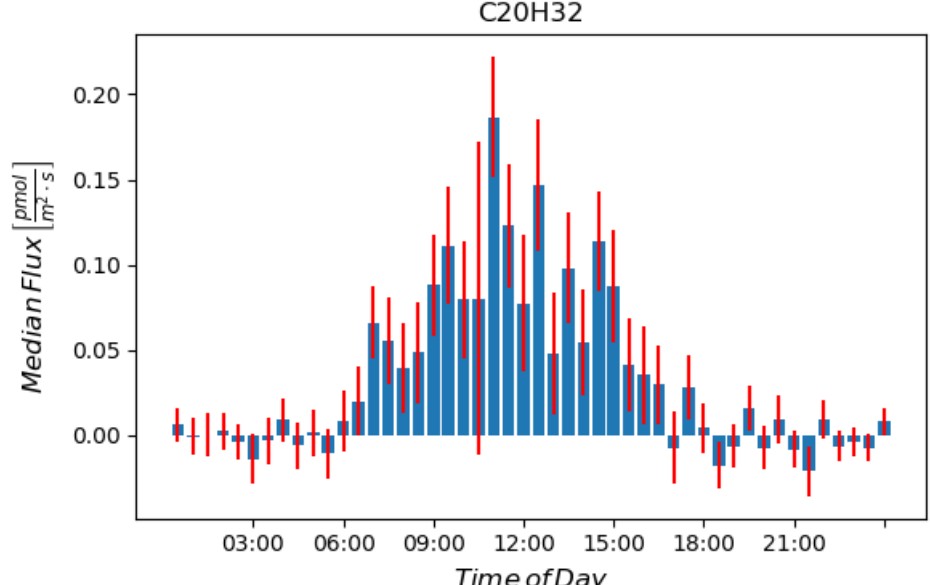

**Figure 8: Diurnal cycle of diterpene emissions. Red lines indicate the random error as described by Finkelstein and Sims (2001) divided by the square root of the number of averaged samples.**

The role of diterpenes in new particle formation and early growth has not yet been thoroughly studied. Atmospheric degradation of monoterpenes ($C_{10}$) generates accretion products composed of the carbon backbone of two $C_{10}$ -$RO_2$ reactants (Berndt et al., 2018a; Berndt et al., 2018b). As described in the introduction, these are contributing substantially to new particle formation. Ambient measurements report concentrations of $C_{16-20}$ HOM in the order of $10^{-2}$ to $10^{-3}$ μg/m³ (Mohr et al., 2017). A substantial fraction of these HOM could also originate directly from diterpene ($C_{20}H_{32}$) oxidation. Giving an estimate on the
diterpene emissions could help to investigate their importance for aerosol formation and necessitate further laboratory and field studies on their role in the initial steps as well as subsequent growth into CCN sizes.

**5 Conclusions**

Accurate quantification of concentrations and fluxes of VOC and SVOC is a crucial component for understanding the formation and evolution of SOA. While conventional methods for measuring VOC composition based on GC-MS sampling
are typically available during atmospheric field measurement campaigns, a critical research need is the quantification of concentrations as well as fluxes of all VOCs and SVOCs closely linked to SOA formation. Fluxes of BVOC and their oxidation products exhibiting reduced volatility such as SVOC or low volatile organic compounds can only be measured with rather short inlets to avoid wall losses during sampling. Here we explored the possibility of using a PTR3 instrument on top of the

SMEAR II tower in Hyytiälä, Finland. We recorded data at 36 m above a forested ecosystem dominated by terpenoid emitters.
We designed and tested a virtually wall less inlet that allows undisturbed gas sampling approximately 4 m away from the tower structure.

At the beginning of the growing season during several warm days in May 2016 we recorded isoprene and monoterpene fluxes supporting previous results measured at the same site. In addition, we report sesquiterpene emission fluxes as low as single digit picomol/m²/s. For the first time, we present emission fluxes of sesquiterpene ozonolysis products ranging from 0.2 to 2
picomol/m²/s. With this setup most suitable for direct eddy covariance flux measurements we were able to detect diterpene emissions lower than 0.15 picomol/m²/s. With the low flux signal to noise ratio achieved with the new PTR3, fast processes can be tracked virtually in real time and clear diurnal patterns can now be studied even for smallest emission rates. Ogive analysis suggests that the new inlet design allows an almost contact free transfer of sample air to the PTR3 over several meters.

## Appendix A

**Humidity dependent in field calibration**

In field calibrations were performed regularly by dynamic dilution of a gas standard (Apel Riemer) containing known amounts of different VOC in dry and humidified synthetic air. Figure A1 shows calibration results for α-pinene at two different days.

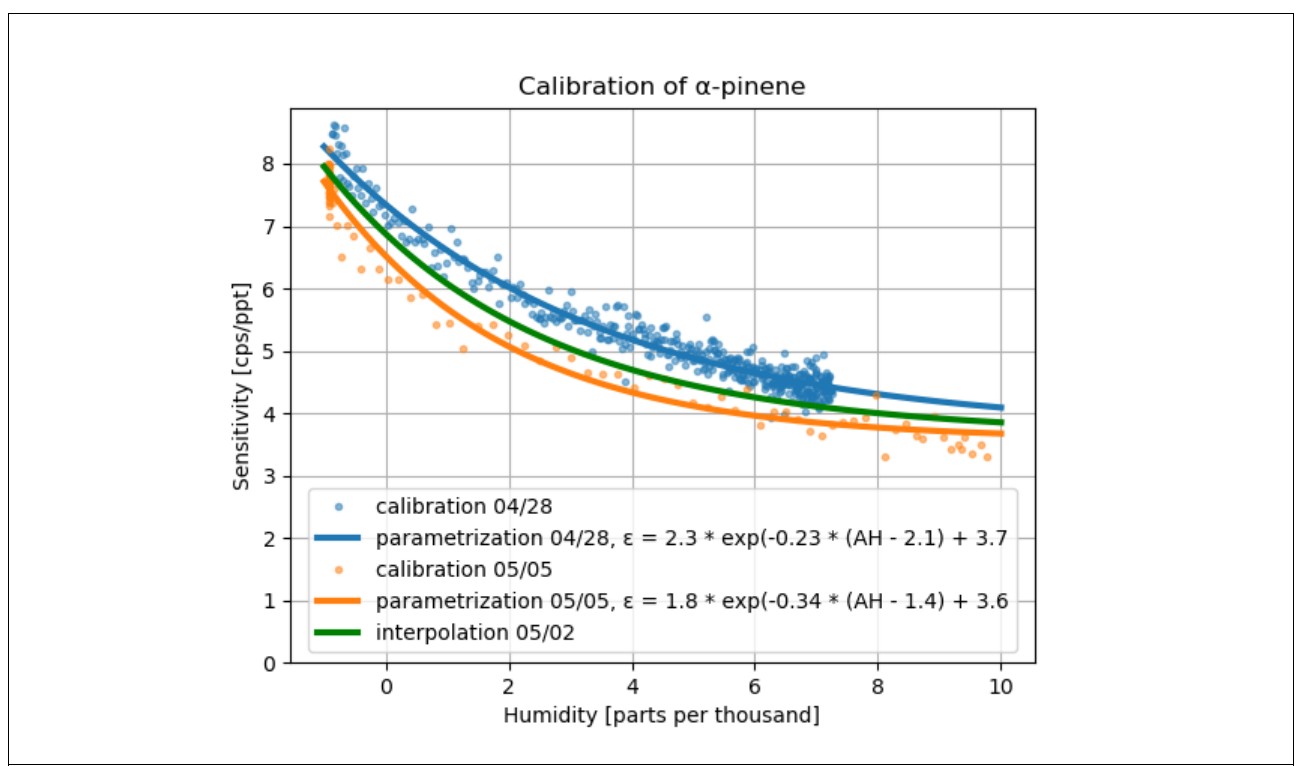

**Figure A1: α-pinene calibrations (blue and orange dots) performed at two different days show two parametrization curves (blue and orange lines). The models for the parametrizations are shown in the legend with sensitivity ε in cps/pptv and absolute humidity (AH) in parts per thousand. Interpolated parametrization curves were used for those**

**days in between the two calibrations (May 5th as an example, green line). Negative humidity values are an uncorrected measurement offset and do not influence our relative parametrization.**

**Fast humidity signal from tracer ($N_2H^+$):**

The recorded ion signal $N_2H^+$ was fitted to the slower infrared gas analyzer signal to derive a fast humidity trace. No lag time correction was applied, since the air measured by the IRGA is directly sampled at the inlet of the PTR3. The advantage of this method is, that the humidity signal is affected by inlet line delays and frequency damping in the same way as the other concentration signals used for eddy covariance analysis. Exemplary results of the method are shown in figure A2. The left panel shows the parametrization of humidity via $N_2H^+$. Data were acquired directly during the flux measurements by comparing
with synchronized IRGA measurements. On the right panel the resulting humidity trace based on $N_2H^+$ is presented. The trace is averaged to 1Hz to match the IRGA sampling rate and shows the noise created by this approach. While this looks worse than the IRGA signal at a first glance, the additional noise is uncorrelated to the vertical wind component, while fast, real humidity fluctuations will be captured better by the higher time resolution of the PTR3. Strong curvature in the humidity dependent sensitivity can however cause a slight shift in average sensitivity when using noisier humidity signals.

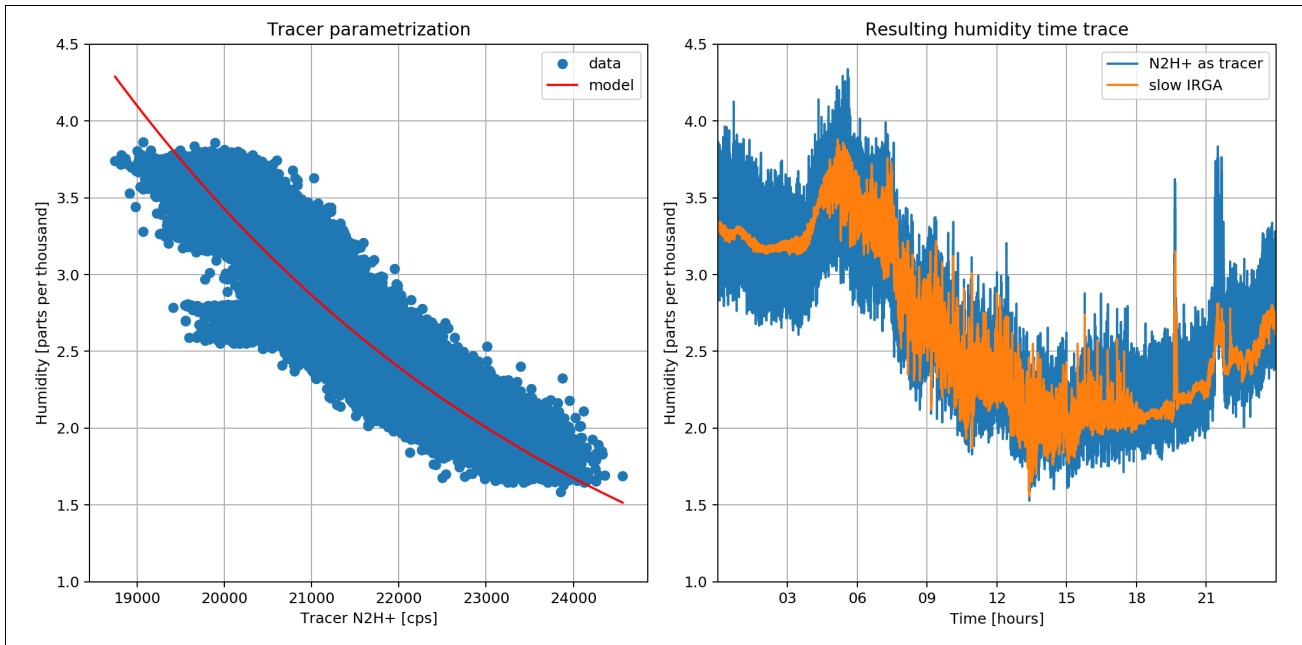

**Figure A2: Left panel: Recorded $N_2H^+$ ion signal as a function of IRGA humidity (symbols) and model fit (red line); right panel: resulting humidity trace derived from fast recorded humidity proxy ion signal $N_2H^+$ compared to slow infrared gas analyzer (IRGA) data.**


We chose $N_2H^+$ over $H_3O^+(H_2O)$ as humidity tracer, since it shows better correlation. The recorded primary ion distribution in the PTR3 is less dependent on humidity than in previous instruments as demonstrated in figure A3.

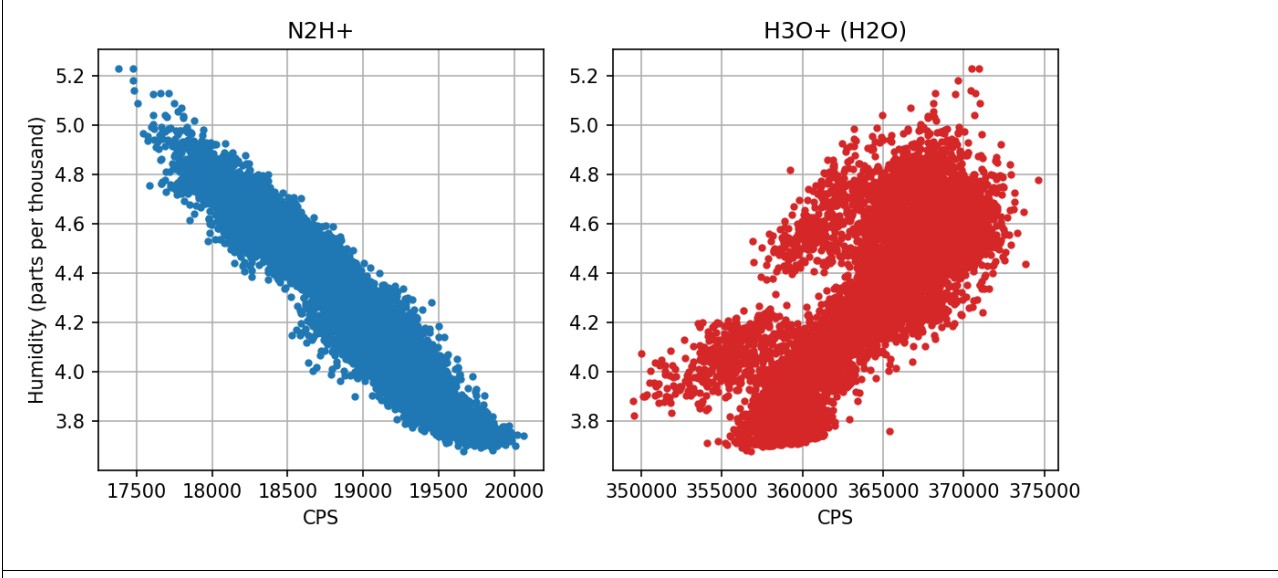

**Figure A3: Comparison of $N_2H^+$ and $H_3O^+(H_2O)$ ion signals as a function of IRGA humidity during May 8th**

For our measurement height, the contribution of high frequencies to the total flux is not as big as above smooth terrain like grassland, so the reconstruction of the fast humidity tracer was not strictly necessary. However, to demonstrate the applicability of our method and in order to show the negative effect of using a slow humidity signal for sample-by-sample calibration, we

analyzed one day of the dataset using three different humidity signals for calibration: $N_2H^+$ derived, unprocessed IRGA and low pass filtered IRGA signal (0.01Hz).

Isoprene has a moderate humidity dependence in the PTR3 and shows a high emission velocity. As shown in figure A4, it is almost unaffected by the modulation effect of water vapor flux. If the effect is completely ignored and a very slow humidity signal is used for calibration (smoothed IRGA), we observe about 5% deviation compared to using either the $N_2H^+$ derived or

the direct IRGA humidity time trace. Methanol ($CH_4O$; protonated exact mass 33.034 Da) is highly humidity dependent in the PTR3 and has a lower emission velocity. Therefore, changes in sensitivity due to water vapor emission mask the flux signal and create artificial deposition when the calibration is not done with adequate time resolution as demonstrated by the smoothed, slow humidity signal. The slightly lower flux resulting from using the $N_2H^+$ derived humidity signal is resulting from the curvature of the humidity dependence, which distorts the noise distribution causing a slightly higher calculated sensitivity on

average.

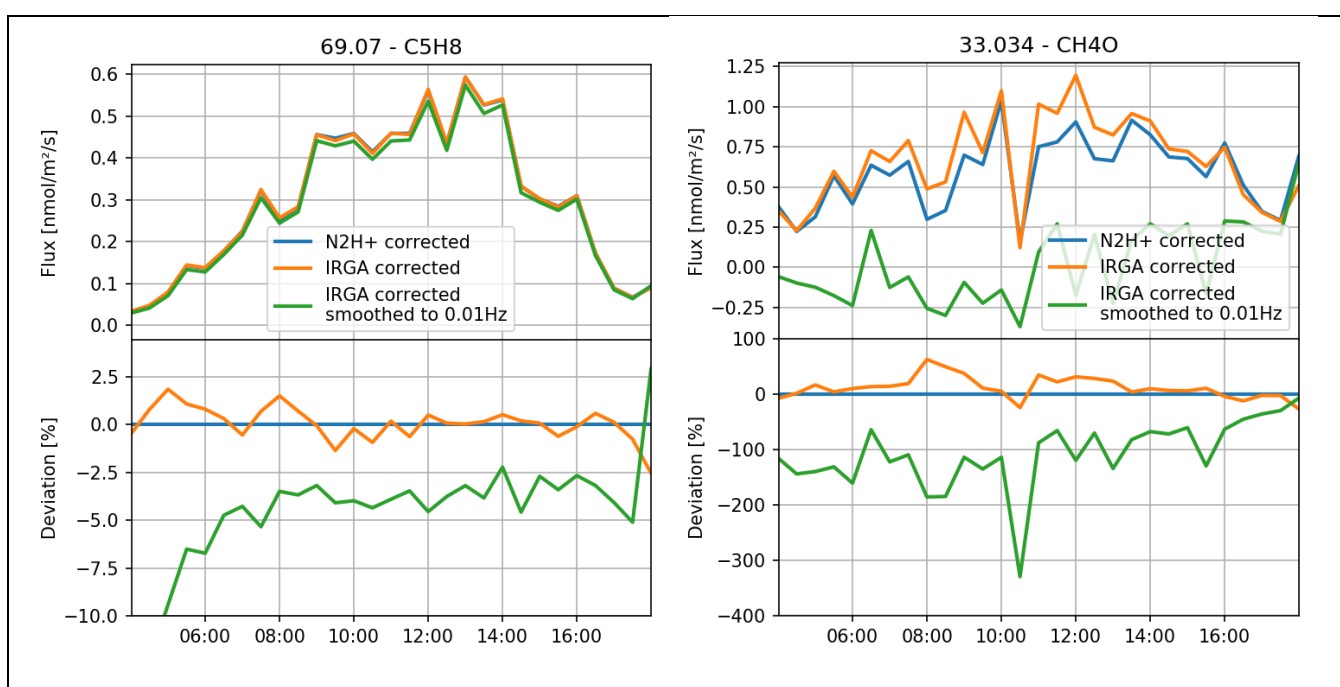

**Figure A4: Isoprene (C₅H₈; left) and methanol (CH₄O; right) fluxes analyzed and compared with three different humidity signals for calibration.**

## Appendix B

**Comparison of diurnal average fluxes and concentrations**

For easy comparability, we present diurnal average data for isoprene, sesquiterpene, the potential β-caryophyllene oxidation products $C_{15}H_{24}O_3$ and diterpene in an alternative representation to the time traces presented in the results. Data are processed as in the section of diterpene, the error bars this time are including uncertainty resulting from day-to-day variability during our averaging period.


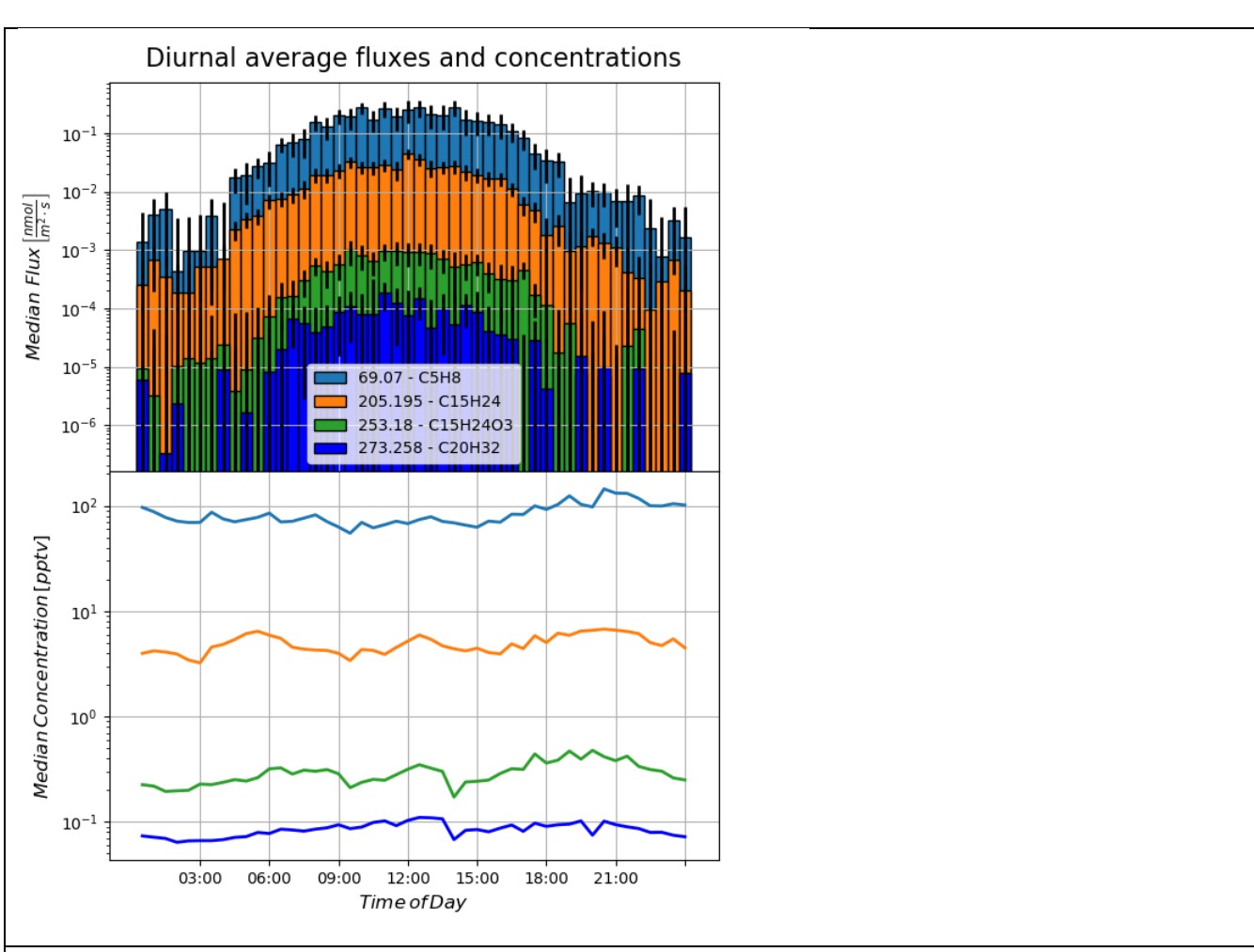

**Figure B1: Diurnal fluxes and concentrations of isoprene, sesquiterpene, β-caryophyllene oxidation products and diterpene.**

**Power spectra**

Power spectra for vertical wind, virtual temperature and scalars measured by PTR3 were calculated over the period from May 3rd to May 8th. The traces were scaled by the variance in their signals for better comparability and are presented in figure B2. In contrast to the traces acquired by the sonic anemometer, the PTR3 traces contain a high contribution of white-noise. This contribution depends on the ion count rate of the measured compound. Since this noise is uncorrelated to the vertical wind component, no contribution to covariance and thus flux is expected. Errors resulting from this noise can however not be completely excluded.

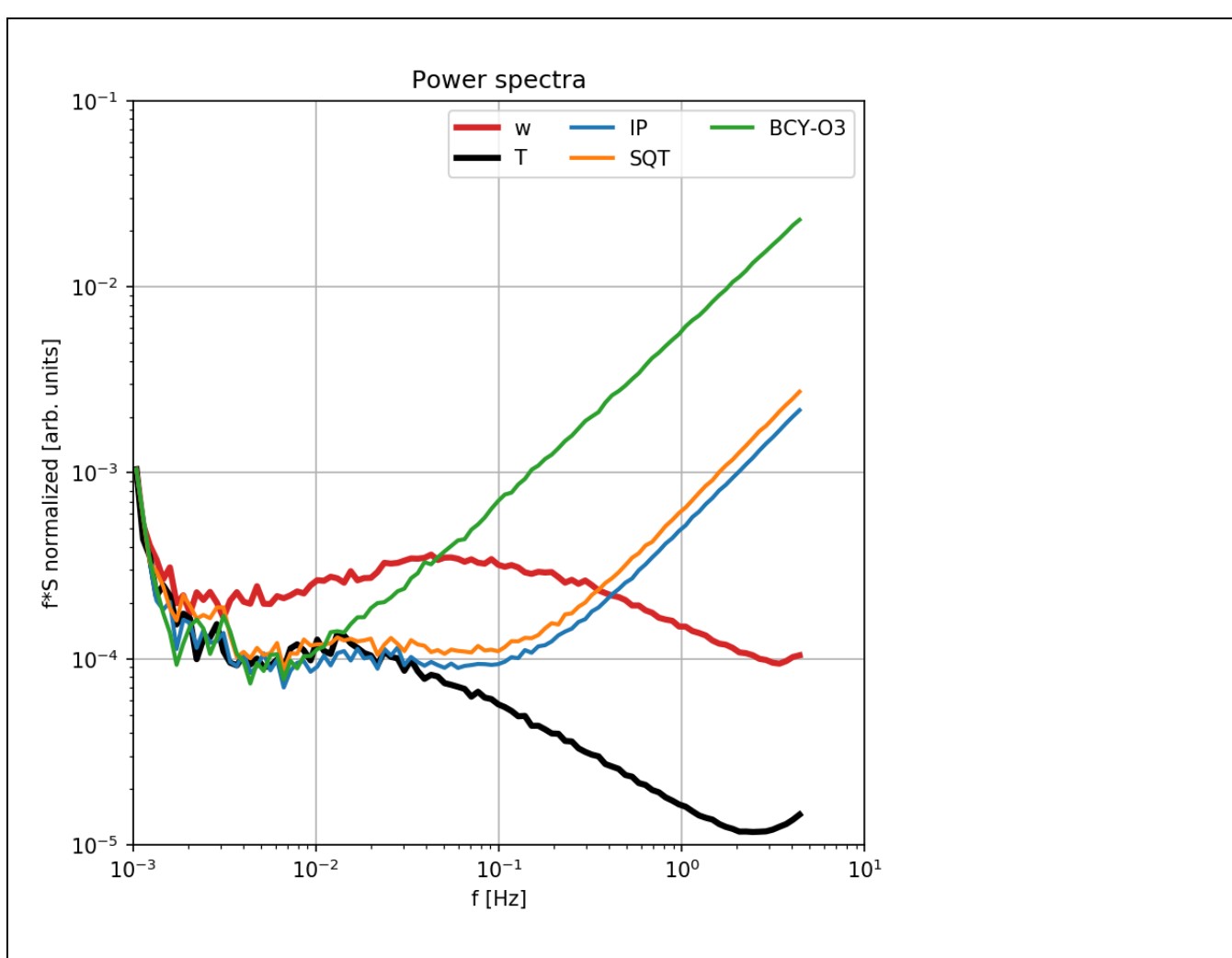

**Figure B2: Power spectra of vertical wind (w), virtual temperature (T), isoprene (IP), sesquiterpene (SQT) and β-caryophyllene oxidation products (BCY-O3)**

## Appendix C

**β-caryophyllene ozonolysis product yield determined in flow reactor measurements:**

PTR3 eddy covariance flux measurements revealed an ion signal of protonated $C_{15}H_{24}O_3$, which we attributed to sesquiterpene ozonolysis products. β-caryophyllene is one of the most prominently emitted sesquiterpenes in Hyttiälä (Hakola 2006, Rinne 2007, Helen 2018). β-caryophyllene reacts very fast with ozone (Richters 2015), so it is presumable the predominant precursor

of $C_{15}H_{24}O_3$. To support this assumption and quantitatively relate the amount measured with PTR3 to the reacted precursor, we performed ozonolysis experiments with the flow reactor in our laboratory. A reaction time of 12 seconds without substantial wall contact is achieved in the Innsbruck flow reactor, which is similar as described in Hansel et al. (2018).

We could add up to 130 ppbv ozone (utilizing an UVP Ozone Generator from Analytik Jena) to a laminar flow of purified air (33 SLM). 2 SLM purified air with 2 ppbv of β-caryophyllene was injected via four impinging jets further down-streams

initiating quick local mixing. At the end of the flow reactor ozone as well as β-caryophyllene and corresponding ozonolysis products were monitored with an ozone monitor (Thermo Environmental Instruments 49 C) and a PTR3, respectively. When switching on ozone β-caryophyllene is partly consumed during the reaction time of 9.4 seconds. More than 40 ion signals corresponding to ozonolysis products could be measured with PTR3. On a carbon basis we detected at least 82±17% of the amount of reacted β-caryophyllene with PTR3. The ten most prominent product ion signals are shown in figure B1, which

correspond to 67±14% of total reacted β-caryophyllene.

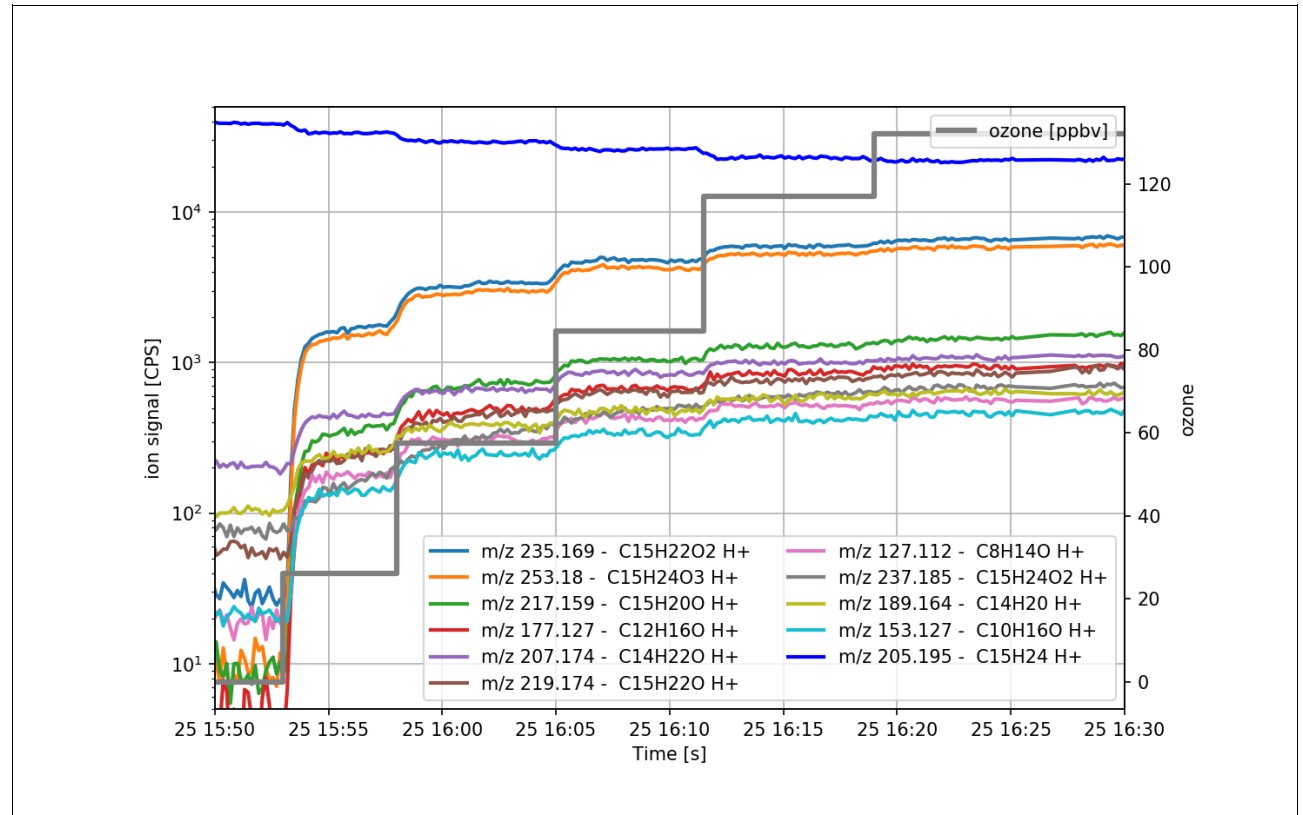

**Figure C1: Ozonolysis products of β-caryophyllene when ozone is increased stepwise as a function of time (thick grey line). The dark blue signal is the concentration of β-caryophyllene, which is decreasing to a new steady state at each time the ozone concentration is increased. The products of the reaction are increasing proportionally to the reacted β-caryophyllene.**

The three most prominent peaks are $C_{15}H_{24}O_3 \cdot H^+$ and two corresponding fragment ions losing one or two $H_2O$ molecules have been recorded. The intensity of the fragment ions was increased when collision induced dissociation experiments were

performed. Yields were calculated from the quotient of product concentration and the reacted β-caryophyllene concentration. The PTR3 sensitivity for β-caryophyllene was calibrated. Sensitivities for product ions were assumed to be similar to ketones which have highest ionization efficiencies (upper estimate). The yield of $C_{15}H_{24}O_3 \cdot H^+$ was 23±5%, the two fragments $C_{15}H_{22}O_2 \cdot H^+$ and $C_{15}H_{20}O \cdot H^+$ had a yield of 20±4% and 5.2±1%, respectively.

**Data Availability**

Processed data of eddy covariance flux and concentrations are provided online by Fischer et al., (2021) at https://hdl.handle.net/20.500.11756/d025b0bb. Data recorded by the SMEAR II station can be found at https://smear.avaa.csc.fi/. The scripts used to process raw TOF spectral data are hosted at https://github.com/lukasfischer83/TOF-Tracer and https://github.com/lukasfischer83/peakFit . The eddy covariance flux
routines are hosted on https://git.uibk.ac.at/acinn/apc/innflux.

**Author Contribution**

LF, MB and EC performed the field measurements. WS assisted LF during the flow reactor measurements. MS provided the flux analysis routines. LF, MB, MG, TGK and AH designed the flux measurement approach. AH, TP and MK acquired funding and supervised the project. LF analyzed the flux and concentration data and drafted the paper. All authors read the paper and
provided feedback that led to improvements.

**Competing interests**

The authors declare that they have no conflict of interest.

**Financial support**

This research has been supported by the Austrian Research Promotion Agency (FFG) under project no. 846050. The field
campaign in Hyytiälä was financially supported by ACTRIS-2 TNA project: "Emissions of BVOC and their oxidation products contributing to New Particle Formation (BVOC-NPF)".

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
