# Peer review of "First Eddy Covariance Flux Measurements of Semi Volatile Organic Compounds with the PTR3-TOF-MS"

_Atmospheric Measurement Techniques, 2021_

## Referee Comment (RC2)

Review of "First Eddy Covariance Flux Measurements of Semi Volatile Organic Compounds with the PTR3-TOF-MS", Fischer et al., AMT (2021)

This paper presents observations of concentrations and EC fluxes of several biogenic VOC above a boreal forest. Measurements are acquired with a new instrument that has much higher sensitivity than its predecessors. They also use a novel inlet design that minimizes contact of sample gas with instrument surfaces. Measurements from a 3-week period in the spring are analyzed to demonstrate instrument performance and highlight the potential of the new capabilities. The paper is generally well written and the number and style of figures is appropriate. Publication is recommended after consideration of the following, mostly minor, critiques.

General Comments

Sections 4.5 and 4.6 deal with data quality and uncertainties. Possibly it makes more sense to present these before the discussion of specific fluxes in Sect. 4.1-4.4.

Specific Comments

L87: It is stated that the inlet mounting was meant to minimize flow distortion. And later on L145, it is stated that the "anemometer deviation" is < 2 cm/s. What does this mean, precisely, and how was it determined? Were any tests carried out to confirm that flow distortion does not influence fluxes (e.g., comparing heat and momentum spectra/cospectra with the inlet flow on/off)? Also, is there potential systematic bias associated with such large sensor separation?

L119 – 121: This sentence might be more appropriate for the introduction.

Figure 2: An actual photograph of the inlet/sonic setup, if available, would be helpful.

L274 – 278 and 295 – 303: This is stated in the introduction and does not need to be reiterated here. Deletion or shortening suggested.

L372: Is this a lower limit for the Bcary emission rate, since it does not account for deposition of the oxidation products?

L388 – 392 – all of this description is more appropriate for the figure caption. Discussion here should focus on what the figure means, not the visual elements.

Figure 6: It would be helpful to see this type of plot for all the fluxes (and maybe diel average mixing ratio, too?).

L423 – 426: The exact same sentences, or nearly so, appear earlier in the manuscript (295 – 303).

Sect. 4.5: It would also be instructive to see the power spectra, either in the main text or supplement.

Figure 8: Would it be possible to add the w' – temperature lag correlations for comparison? These lag peaks seem very wide.

Technical Comments

L33: "critical"

L51: "predicted that"

Figure 1: I am somewhat confused by the cartoons. What is the "measurement container" box meant to show? And the "inlet box", which seems to appear opposite of the stated inlet orientation?

L102: "empirical"

L105: "expected"

L149: What is the brand or manufacturer of the blower?

L351: it seems like pmol would be a more convenient unit here.

Figure 4: why are concentrations shown in cps instead of pptv or similar mixing ratio units?

L421: "of the number of averaged samples"

---

## Author Comment (AC1)

**Reply on RC #1**

**General comments**

"First Eddy Covariance Flux Measurements of Semi Volatile Organic Compounds with the PTR3-TOF-MS" highlights the utility of the PTR3-TOF-MS in eddy covariance studies to observe the forest-atmosphere exchange of terpenes and sesquiterpene oxidation products with the implication of extending this application to other molecules. It adds many useful contributions to the current state of BVOC flux literature including an inlet design for limiting attenuation of low volatility compounds, direct flux measurements of compounds in low abundance like sesquiterpenes and diterpenes, the first eddy covariance measurements of diterpenes, and a novel analysis using direct measurements of oxidized products and precursors to obtain speciated flux. This paper should be accepted after mostly minor revisions listed below. These revisions primarily concern clarifications in experimental setup and in flux quality control and assessment that will ultimately aid someone new to eddy covariance that wants to do a flux measurement with their PTR3. I have a less minor set of comments on the authors' treatment in calculating the B-caryophyllene oxidation product (BCYO3) and want to know why they only consider reactions from the canopy top to the sensor rather than through the whole canopy.

**Specific comments**

Line 70: Please list the reference for this 8% value.

*We added the Hari and Kulmala publications as source of this information.*

Line 100-101: How far away was the IRGA from the PTRMS? It is not mentioned how long the instrument bypass is where both the PTRMS and IRGA are sampling. Further, it is not specifically mentioned if the IRGA is in the same trailer as the PTRMS. These points should be clarified.

*The IRGA was mounted directly inside the PTRMS rack and sampled the excess air drawn from the core sampling. Sampling line length was short (<0.5m). We added clarification to the "Wind and humidity data" section.*

Line 103: What drift tube pressure was used in this study? This would be a helpful value.

*We operated the instrument at 76 mbar. This was added to the text.*

Line 107: Please state the recent you use m/z 100.

*The mass dependence of the TOF transmission is dominated by the duty cycle of the TOF extraction and can be approximated by a square root function. The reference point where the transmission is assumed to be 1 can be arbitrarily chosen. In our case we use m/z 100 since correction factors for our compounds of interest are close to unity. The resulting calculated concentrations are however of course not influenced by this choice. We added a short note to the manuscript.*

Line 117-118: what were the other masses used in the mass calibration? Mentioning this would help show the points of calibration for your mass range and provide a starting point for others.

*Only the described two compounds with m/z 21 (measurable isotope of primary ion H3O+) and m/z 371.102 (protonated siloxane) were used to fix the mass scale. The mass and sum formula of the siloxane was added for convenience.*

Lines 119-121: You should add a line about what the gap is. Is it that nitrate and iodide can detect OVOC at have high sensitivities that PTR previously could not, but those methods can't detect hydrocarbons?

*In order to increase the sensitivity of a CIMS the number of reactive collisions between precursor ions and the compounds have to be increased. There is a problem with secondary reactions for high abundant compounds (hydrocarbons in the atmosphere). If the pressure and the reaction time are too high precursor ions are depleted. PTR3 uses a typical pressure of 76 mbar and a reaction time of 3 ms as a compromise. This allows bridging the gap to detect abundant hydrocarbons (ppbv range) and less abundant OVOC (< pptv range) in the atmosphere. The gap was*

*described in more detail. Due to remarks from reviewer #2 the paragraph was moved to the introduction, where it fits better.*

Line 200: How often were field calibrations using the cylinders performed? I understand that the humidity correction is at 10 Hz but it's not mentioned how often calibration curves are ran.

*We calibrated every couple of days, preferably during bad flux conditions (at night or during rain). We changed the sentence accordingly.*

It's helpful for other readers to get an idea for what frequency of calibration works well for a PTR3 eddy covariance study.

*Calibrations were done based on stability of the instrument. A corresponding sentence was added.*

Further, is there a notable difference at this height in calculated flux if you apply a 1 Hz humidity correction as compared to the 10 Hz one? It would be helpful to show the difference and the utility in this N2H+ method.

*We used the 10 Hz humidity correction utilizing N2H+ as proxy signal in order to demonstrate the capability of this method. We did also re-analyze one day of flux data and added a section with the results below to appendix figure A4 to show the effect of slow humidity signals. A 1 Hz correction would be sufficient at our measurement height for the compounds reported in in this paper, as can be seen by the almost indistinguishable signals in the case of isoprene. In the case of methanol, the errors resulting from slow humidity signals for calibration are more pronounced due to its strong humidity dependence and its low emission velocity. In the case of an extremely slow humidity signal (low pass filtered to 0.01Hz) emission is even falsely reported as deposition. The slightly reduced flux of our $N_2H^+$ corrected trace is probably resulting from the additional noise from the mass spectrometer causing slightly overestimated sensitivity on average due to the curvature of its humidity dependence.*

[Figure]

Line 208: is this correlation made by averaging the N2H+ data down to 1 Hz? Can you present the correlation coefficient for N2H+ against humidity compared to that of H3O+(H2O)? Since the latter is more frequently used, showing the better correlation can help strengthen your finding.

*Yes, the N2H+ is averaged down to 1Hz to generate the scatter plot of N2H+ vs. humidity. The relevant drifts determining the parametrization are fairly slow (minutes to hours) compared to the sampling rate, so this is justified.*

*The correlation between N2H+ and humidity was more immune to instrumental drifts than H3O+, since it is created after the ionization. A paragraph and the following figure were added to appendix A.*

[Figure]

Line 221-222: Does "treated similar to acetonitrile" mean you used the same calibration factor as acetonitrile for the remaining compounds? If so, you should just say that.

*Yes, we changed the text accordingly.*

Line 224: Since you are mentioning the effects of water you should include what the effect of fluctuations in air density from water vapor fluctuations air as in the WPL correction from Webb, et al. 1980. I assume it would have a small effect since your measured mixing ratios are small, but it would be helpful to just mention this as requiring consideration for flux measurements and the net effect:

*Since the calibrations of the PTR were done with a fixed dilution rate of a gas standard with dry air, adding humidity after the mass flow controller of the dry air, the humidity dependent calibration already includes the dilution effect caused by the additional water vapor. So, we are directly measuring dry sample concentration, which does not require further WPL correction. This was shortly explained in the text.*

Webb, E. K., Pearman, G. I., & Leuning, R. (1980). Correction of flux measurements for density effects due to heat and water vapour transfer. Quarterly Journal of the Royal Meteorological Society, 106, 85–100. https://doi.org/10.1002/qj.49710644707

Line 274: This site is not composed of broadleaf trees so this sentence does not read as relevant to the study as written. I understand this is meant to generally say that this instrument is useful in measuring an abundant NMHC but it might be more helpful for the point of this study to just merge the first few sentences of this paragraph and remove the broadleaf statement. If you do want to keep it then you should mention if you expect high isoprene emissions from the scots pine at your site or know where the isoprene comes from.

*Reviewer #2 mentioned similar concerns and the section was shortened.*

Line 281-282: Was this determined using measured fluxes and a calculation of a dilution rate and reaction rate from estimated OH? Or is this just a suggestion as to why this is happening? Would the authors be able to show that at relevant OH and dilution the source and sink are close? Or perhaps this has been shown in previous studies at this site?

*This explanation was meant to illustrate the benefit of measuring flux in contrast to concentrations, since concentrations reflect the current steady state resulting from changing sources and sinks. No calculations were made to support our claims, which should just draw a rough picture of what is happening in the plots in figure 3. The text was modified and speculative statements were avoided.*

Line 292-293: I am a little confused about the importance of the start of the growing season on isoprene emissions and it should be clarified. Is this sentence implying that there is extra leaf mass from new needles during the growing season that contributes to isoprene emissions or is there an initiated biochemical pathway that solely happens during the growing season that is not temperature dependent? Are you saying that deciduous leaf area increases from the generation of new springtime leaves? Is there enough deciduous leaf area in this site? Or is it just mentioning it got warmer earlier?

*We now give a more differentiated discussion of the deviation of a factor of four from previously reported values. Indeed, our measurements took place during an unusually warm and sunny period. Using the basic emission model for isoprene from Guenther 1997, we expect 200% emission compared to Schallhart 2017 due to differences in PAR and temperature. We can however not fully explain the remaining factor of two, but we expect an earlier increase in foliar density from understory and scattered decidious species as well as earlier decrease in snow cover (unfortunately no smart smear data available for 2013) which further increases isoprene emissions.*

*Due to fragmentation, MBO is detected on the same exact mass as isoprene. Therefore, we cannot exclude contributions of MBO to the detected signal at m/z 69.070, which we termed isoprene.*

*References to Guenther (1997) and Aalto et al. (2014) were added.*

Line 331-332: Is the "known temperature dependence" taken from the fit of the data in Rinne et al. (2007)? Or do you normalize using a Beta factor of 0.09 /K and then assume a same basal emission rate across studies and seasons? It should be mentioned where this is taken.

*We used their normalized emission potential and temperature dependence of m/z 137. The text has been changed accordingly. Additionally, a reference to Taipale 2011 was added, who measured at the same site and time of the year and is in good agreement with our reported emissions as well.*

Line 318-320: Is the aforementioned monoterpene source the sawmill that is a source for a lot of signal enhancement? The strong downward flux on 05/08 is very interesting. Do you have any ideas on what would be causing a monoterpene deposition? There probably are not any soil processes strong enough to drive this, right? Horizontal inhomogeneities would not contribute to an error in turbulent vertical flux since your vertical flux would then just be from the average of whatever monoterpene is sampled. Could you have outsourced, very reactive monoterpenes that are horizontally transported then are chemically lost below the sensor when vertically processed within your footprint? Or is it just that canopy MT is much lower below the sensor than the outsourced MT plume at the sensor and you are just seeing eddy diffusivity? There should be a some hypothesis as to why this is happening since this is an anomaly.

*The monoterpene source in the example on May 8th is definitely not the sawmill, it comes from a different direction (south-west instead of south-east). We have no indication on the distance of the source either. We investigated the fragmentation pattern in comparison to α-pinene and found no significant change during the deposition events. Unfortunately, this can still not exclude the theory of a highly reactive monoterpene, since it is not uncommon for different monoterpenes to share the same fragmentation pattern. A definitive answer could only be given by measurements of the monoterpene composition by GC-MS. We share/favor the hypothesis that canopy MT is lower than the incoming "plume" and therefore we just see diffusion of a very sharp, nearby source into the canopy where horizontal transport causes some dilution before it is re-emitted.*

Figure 4: What is the frequency of data used in these histograms? 10 Hz data from a 30-minute period?

*Exactly, we adapted the figure caption.*

Also are you able to report concentrations of monoterpenes and isoprene in humidty-corrected pptv and does the figure look any different when you do that? Monoterpene concentrations in pptv are not presented anywhere in this manuscript. Does the time series look odd because of the horizontal imhomogeneities discussed? It should be mentioned why a time series of monoterpene concentrations is not presented since it would have been helpful to get an idea of the impact of these downward fluxes as well as to infer relative chemical and dilution lifetimes as you do for isoprene. It would also have been helpful to show the relative concentrations of isoprene and monoterpene just from a terpene abundance and sourcing point of view. I understand the purpose of this paper is not to provide a full characterization of the site but to show the utility of the method for eddy covariance. However, a more direct statement of why a monoterpene concentration time series or even listed averaged monoterpene concentrations in the text are not reported should be made.

*Histograms were recalculated using humidity calibrated time signals, no qualitative changes can be found. The new results are used for the plots now for consistency with other data presented in the paper.*

*The monoterpene time series was added to figure 4. With the naked eye, the concentrations do not look odd during the deposition events, only slightly higher fluctuations can be observed. This must be however partly attributed to the averaging of 30 min by the flux data processing. In full 10 Hz the directional variations are of course more pronounced, but this information is already better presented through the histograms. Concentration spikes originating from the*

*sawmill on 5/12 and 5/14 show some influence on the flux data, but they were observed mostly during the night while flux conditions were not met.*

Line 365: One of the highest sesquiterpene rates at this site? From Scots Pine? In general? Please specify.

*Measured at the same site. Clarification was added.*

Line 376: Is FBCYO3,tower a calculated or directly measured value at the tower? If it is calculated then this is not clear.

*It was the flux value measured by the PTR3 at the tower. Anyway, due to modifications in this section, the corresponding sentence was removed.*

Line 381-382: Is this yield from the lab study dependent on reaction time? Would there be a further correction of yield applied in the Line 375 equation based on residence time?

*We calculated the yield by the fraction of the observed production of the product divided by the observed consumption of the precursor, both were measured during the flow reactor study. So further reactions of the product will in theory lower the observed fraction for longer reaction times. While we did not study the dependence of this fraction on the reaction time ourselves, it will still be quite constant (within the relevant time scale for the calculations of several minutes), since the atmospheric lifetime of the product BCYO3 is in the order of hours (Winterhalter et al. (2009)). The lifetime of BCYO3 was mentioned in line 368.*

Line 383: Should remind that H = 15 m. You should also present your ranges in u* since they are applied in the residence time calculation in Figure 5.

*The new discussion of our BCYO3 observations do not use residence times anymore, u* is presented as a qualitative hint instead (see changes due to the next comment below).*

Further, why are you only considering the reaction time from the canopy top to the sensor and not from within the canopy to the sensor (as in Fulgham, et al. 2019 and Vermeuel et al. 2021)? It is likely that a sizeable fraction of your BCARYO3 was made within the canopy. I believe in the cited text (Karl, et al. 2018) the authors use the distance traveled from the injection site which is where the compound emanates from (in their case a point source). Daytime parcel residence times can be in the range of minutes (Fuentes, et al. 2015) under high turbulence conditions (above canopy u* = 0.65 m/s) when emanating from the understory which is commonly where sesquiterpenes originate from. Zhou, et al. (2013) showed through a 1D vertical model at the SMEAR II site a that 70% of SQT reacts within the canopy and only ~29% escapes from the canopy. Are you only considering the reaction of that 29%?

*As pointed out by reviewer #1, our oversimplified estimate neglected in-canopy chemistry producing additional BCARYO3. In-canopy residence time could be roughly estimated based on the publications proposed by the reviewer, but also depends on the uncertain exact height profile of sesquiterpene sources. Coherent structures commonly found in complex terrains with rough surfaces like forests further complicate the prediction of parcel residence times as mentioned in the next comment. Uncertain exact composition of sesquiterpene emissions as well as the potential deposition of BCARYO3 within the canopy mentioned by reviewer #2 are further unknowns. Consequently, we decided to reduce our analysis to variables directly derivable from our observations. We now calculate back to BCARY from BCARYO3 using the laboratory ozonolysis yield and discuss the daily fraction of reacted BCARY compared to the measured total sesquiterpenes. The observed temperature dependence and the unexpectedly missing dependence on u* are mentioned but left unexplained. The focus of this work is on the technical aspects of using the PTR3 with the optimized inlet for improved transmission of semivolatiles, these calculations were only intended as inspiration for others profiting further from similar highly time resolved observations (in order to refine emission models, ...). Due to the short measurement period, our measurements can only show a snapshot and further speculative analysis could probably not contribute much to existing emission models and inventories.*

Fulgham, S. R., Brophy, P., Link, M., Ortega, J., Pollack, I., & Farmer, D. K. (2019). Seasonal Flux Measurements over a Colorado Pine Forest Demonstrate a Persistent Source of Organic Acids. ACS Earth and Space Chemistry, 3(9), 2017–2032. research-article. https://doi.org/10.1021/acsearthspacechem.9b00182

Vermeuel, M. P., Cleary, P. A., Desai, A. R., & Bertram, T. H. (2021). Simultaneous Measurements of O3 and HCOOH vertical fluxes indicate rapid in-canopy terpene chemistry enhances O3 removal over mixed temperate forests. Geophys. Res. Lett. https://doi.org/10.1029/2020GL090996

Fuentes, J. D., Wang, D., Bowling, D. R., Potosnak, M., Monson, R. K., Goliff, W. S., & Stockwell, W. R. (2007). Biogenic hydrocarbon chemistry within and above a mixed deciduous forest. Journal of Atmospheric Chemistry, 56(2), 165–185. https://doi.org/10.1007/s10874-006-9048-4

Zhou, P., Ganzeveld, L., Taipale, D., Rannik, Ü., Rantala, P., Petteri Rissanen, M., et al. (2017). Boreal forest BVOC exchange: Emissions versus in-canopy sinks. Atmospheric Chemistry and Physics, 17(23), 14309–14332. https://doi.org/10.5194/acp-17-14309-2017

Line 383-384: It should be at least mentioned that surface layer scaling theory or any type of scaling is not always applicable, especially as you get nearer to the canopy surface and are within the roughness sublayer that is associated with multiple length scales. It is likely that non-local mixing frequently occurs, and the canopy undergoes sweeps and ejections of air parcels where the reaction time is harder to assume. This concept is explained further in Cilfton et al. (2020), Section 4.4. There the authors consider O3 but the idea is the same:

Clifton, O. E., Fiore, A. M., Massman, W. J., Baublitz, C. B., Coyle, M., Emberson, L., et al. (2020). Dry Deposition of Ozone Over Land: Processes, Measurement, and Modeling. Reviews of Geophysics, 58(1). https://doi.org/10.1029/2019RG000670

*While the reviewer is of course pointing out an important issue in the original version of the discussion, this comment is of less relevance to the rewritten parts about SQT and BCARYO3 emissions.*

Line 388-389: How were these percentages chosen for the upper and lower limits?

*The new verion of the manuscript uses no reaction time estimates anymore.*

Further, if you were to pick another common sesquiterpene that you expect to see and factor that error into your k value in the equation on line 375, would it be beyond your listed range in error? Are there any other sesquiterpenes at all from Scots Pines you would expect to emit? You have the reference for speciated SQT dominated by beta-caryophyllene from a July study, but do you expect there may be a seasonal dependence in speciation since you measure in spring? You should mention that there should be no seasonal dependence, if so.

*The atmospheric lifetimes of most other sesquiterpenes reported at this site are much longer than those of beta-caryophyllene (Hakola, H., Hellén, H., Hemmilä, M., Rinne, J. and Kulmala, M.: In situ measurements of volatile organic compounds in a boreal forest, Atmos. Chem. Phys., 12(23), 11665–11678, doi:10.5194/acp-12-11665-2012, 2012.), and could hardly explain the high fluxes of the observed oxidation product. To our knowledge only alpha-humulene shows a similar reactivity. alpha-humulene does not seem to produce $C_{15}H_{24}O_3$ to the same extent, but rather $C_{15}H_{24}O_4$ which we do not observe in similar quantities in Hyytiälä (see Beck, M., Winterhalter, R., Herrmann, F. and Moortgat, G. K.: The gas-phase ozonolysis of α-humulene, in Physical Chemistry Chemical Physics, vol. 13, pp. 10970–11001., 2011.)*

Is your C15H24O3 in lab only detected from ozonolysis of beta caryophyllene and no other likely sesquiterpenes? Stating that would be further evidence that the dominant SQT is only B-caryophyllene.

*In the laboratory flow reactor experiments we used pure beta-caryophyllene. Also, the fragmentation pattern of the BCY-O3 product matches the pattern observed in Hyytiälä of $C_{15}H_{24}O_3H^+$ and $C_{15}H_{22}O_2H^+$. This was mentioned in lines 379 and 382. Still ozonolysis of other sesquiterpenes like for example farnesene are known to produce $C_{15}H_{24}O_3$ as well.*

Figure 5: Showing your SQT and SQT oxide flux at night seems misleading since you likely don't pass your flux filters at night and it is hard to generate turbulence. Further, it looks like all data is included for the bottom panel since you get reaction times greater than 50 s which would be the longest reaction time with your u* of 0.3 m/s filter. It should be specified that you do not filter data in this figure, and you should present what the average and range in daytime reaction times are in the text as well as the average and range in daytime reacted BCARY. I believe panel B would be more helpful if it presented the daytime (~10-18 local time) averages rather than the whole day which includes nights of 100% of a small, more uncertain BCARY source reacting away. If you already do only include daytime averages then this should be specified.

*Thank you for the mindful observation, we forgot to add the description of the middle pannel. The average was indeed already calculated during daytime (in our case 8-18h local time) without indication in the text. In the revised manuscript the newly presented reacted fraction and temperature replace the original data in the middle pannel. The averaging is however done the same was and is described better now.*

Lines 410-412: If providing a survey of CIMS methods that can detect pure hydrocarbons at high sensitivity then you should mention benzene CIMS from Lavi, et al. 2018. They report very high sensitivities for terpenes although they might not be able to get down to the LoD of your PTR3.

*Thank you for bringing this new instrument to our attention, we added a reference, (note: the corresponding paragraph was moved to the introduction).*

Lavi, A., Vermeuel, M. P., Novak, G. A., & Bertram, T. H. (2018). The sensitivity of benzene cluster cation chemical ionization mass spectrometry to select biogenic terpenes. Atmospheric Measurement Techniques, 11(6), 3251–3262. https://doi.org/10.5194/amt-11-3251-2018

Lines 413-414: Are the two weeks of flux measurements just the length of the full study or was a portion chosen? Can you mention that this is just the whole study if the former?

*We started acquisition on April 16$^{th}$, but the first week was not analyzed due to bad weather conditions (cold, snowfall) resulting in low VOC concentrations and fluxes. We added more detailed information in the manuscript.*

Line 435-441: While these ogives do look nice, a visual assessment of a semilog plot is a little misleading for readers. The area you are highlighting and comparing to is halfway through your total area. The authors should calculate what they expect the error attributed from inlet dampening and sensor separation should be by calculating a transfer function using Massman et al. 1991 and Moore 1986. If it does not look too messy on the plot it would also be helpful to include the calculated transfer function. Based on your flow rate and wall length it should only help support your claim of no visible dampening. You could also show what the resulting transfer function is when comparing to wT. There should be a more quantitative description to support your statement of little to no attenuation.

*We provide here a version of the figure with linear scaling for discussion. The largest deviation from w'T' is seen on $C_{15}H_{24}O_3$, which is mostly caused by the noise in the high frequency part of the cospectrum, likely due to the low concentration of the compound. However, the important feature of the ogives remains unchanged in both representations: their asymptotic convergence towards 100% at the high end of the spectrum. This asymptotic value is solely determined by scaling them so that they match the contribution of w'T' at the frequency of the maximum of the cospectrum (grey area, around 0.2 f\*z/U). This includes all low frequency contributions up to this frequency. In case of systematic high frequency dampening, corresponding cospectra would decrease faster than w'T' with rising frequency and their scaled integrals (ogives) could not reach 100%. We think that this is a very intuitive way of assessing high frequency dampening by direct comparison to w'T', which does not need models or assumptions in addition to the raw measurements. Further, due to the short measurement period our measured cospectra are still quite noisy as can be seen in figure 7 (old manuscript, now fig. 3), which makes spectral comparison harder. Additional clarification was added to the text.*

*The proposed calculation method of a transfer function of a turbulent flow through a tube by Massman (1991) is assuming a fully developed turbulent flow profile. This is not the case for our inlet concept, which takes advantage of the mostly undisturbed entrance cone formed at the beginning of a tube. The transmission through our tube with such low L/a aspect ratio will be underestimated by Massman (1991) as he describes around equation (12).*

*Concerning sensor separation, the sonic was mounted vertically above the inlet. This configuration is not covered by Moore 1986 who only discusses horizontal lateral and longitudinal sensor separation. The deviation caused by vertical sensor separation with the wind sensor above the scalar sensor is however discussed in Horst et al. 2009 and was reported to cause the least attenuation of fluxes. They present measured attenuation versus the fraction z/z', which is wind sensor height divided by inlet height. In our case this is 15.5m/15m and the expected attenuation seems negligible.*

*Also, it does not seem to be frequency dependent. We added a reference to the "Wind and humidity data" section.*

[Figure]

Massman, W. J. (1991). The attenuation of concentration fluctuations in turbulent flow through a tube. Journal of Geophysical Research, 96(D8), 15269. https://doi.org/10.1029/91jd01514

Moore, C. J. (1986). Frequency Response Corrections for Eddy Correlation Systems. Boundary-Layer Meteorology, 37, 17–35.

Line 441-443: Were the present cospectra also made in the same way the presented ogives were determined? By taking an average of fluxes that passed quality control tests? Or is this a collection of a single, representative averaging period? If the latter, then you should show an average of more than one to be more representative of the study.

*The cospectra are averaged over the same set of samples as the ogives, both filtered as described in the text which cover the whole dataset we analyzed. We used the cospectra with their frequency axis scaled for U from InnFLUX, which are intended for averaging. Clarification was added to the text.*

Figure 7: The purpose of the grey bar should be indicated in the caption well.

*The caption was adjusted to include a description of the grey bar.*

Section 4.6: Uncertainties are presented throughout the text in the respective sections of detected analytes so this section written as is may be repetitive and unnecessary unless the authors provide a quantitative summary of uncertainties for all presented compounds and/or want to explain in more detail their uncertainty method. Either way, the uncertainty method should be explained more at some point in the text (either when it is first introduced or in this uncertainty section) in the case that the reader does not want to go back and read all of Finkelstein and Simms (2001). This could be a few sentences and does not need to be a rewrite of the cited paper. Also, there are some compounds (monoterpenes, diterpenes) where the study uncertainty is not mentioned so they need to be mentioned at some point in the manuscript.

*We moved the sections to the experimental chapter and merged it with the discussion of calibration uncertainty. The method of Finkelstein and Simms (2001) was shorty explained.*

We added the random flux error for the monoterpene discussion. The random errors of the presented diurnal cycle of diterpenes are visually shown in the figure and their calculation is described in the figure caption. We only give a lower estimate of flux and concentration, since we do not have calibrations and must assume ionization at the kinetic limit.

In addition, unless the uncertainties are calculated by using only the cross-covariance as in some of the methods of Langford et al. (2015), then Figure 8 does not seem to fit or at least there is no connection between cross-covariance and uncertainty as presented. It is an excellent figure that highlights the utility of the inlet but adds more to the previous section and should be included there instead and that whole section could become a "spectral analysis" section. The connection should be further explained if it is related to uncertainty.

*We also moved the lag times to a separate section in the experimental chapter as proposed. It is referenced in the uncertainties section, where we added a statement on lag time determination errors related to our inlet concept. We thank the reviewer for the proposed literature, which gives a very comprehensive overview of flux error sources and helped to better structure our discussion.*

Langford, B., Acton, W., Ammann, C., Valach, A., & Nemitz, E. (2015). Eddy-covariance data with low signal-to-noise ratio: Time-lag determination, uncertainties and limit of detection. Atmospheric Measurement Techniques, 8(10), 4197–4213. https://doi.org/10.5194/amt-8-4197-2015

Figure A2: Is the humidity data used on the y axis from the IRGA or the met station? Is the trace from the IRGA on the right time-corrected to match the N2H+ signal or are those just both raw data traces? These should both be acknowledged. This also brings me back to my earlier comment of what is the time delay between the air sampled reaching the PTR3 and the IRGA. You mention potential dampening and time delay, so I am assuming they are far from each other?

*The humidity was measured by the IRGA close to the PTR3, which was already added in more detail to the instrumental section due to the earlier comment. We additionally clarified this in the figure caption too. We did not apply lag time correction since the residence time in the tubing is small compared to the response time of the IRGA itself. A comment about that was added to the text. We quickly checked the apparent lag time between the two signals, the maximum of the cross covariance is only 1-2 seconds apart, which corresponds roughly to the gas exchange time defined by internal optical cell volume of 14.5 ml and our sample gas flow of 0.5 slpm through the IRGA. The minimum software selectable averaging of the IRGA is also 1 second.*

[Figure]

Further, is the N2H+ data on the right noisier solely due to a higher collection rate and averaging both down to 1 Hz makes them match better? Including the correlation coefficient of N2H+ and time-corrected water as I mentioned earlier will help show the utility of N2H+ as a water tracer.

*The $N_2H^+$ derived humidity signal shown in the plot was already averaged to 1Hz for better comparison to the IRGA signal. We added a corresponding note to the text. Its higher noise compared to the IRGA signal is created by the mass spectrometer, not the better capture of eddies, which can be seen at night time, where the N2H+ signal noise stays roughly the same, while the IRGA is much smoother. However statistical noise which is uncorrelated to the vertical wind component will be suppressed by our ensemble averaging time of 30 min, while the improvements in capturing the fast correlated humidity changes will better describe the correct water dependent sensitivity. A short mention was added to the text. For discussion we add a direct comparison of $N_2H^+$ derived humidity vs. IRGA signal, both averaged on this*

[Figure]

*ensemble averaging time, which shows very good correlation.*

Line 509: B-caryophyllene is consumed as in completely gone? Or is that the e-folding lifetime? Under how much ozone? This should be clarified. I mention this because it is a little misleading looking at Figure B1 and its caption would make you think that a function of time meant a function of reaction time. Also, the sesquiterpene is not completely consumed in the figure.

*Text and figure caption were updated to better describe the experiment. β-caryophyllene is only partly consumed, depending on the ozone concentration.*

Figure 1B: There needs to be a little more clarification on what this figure is showing. You are sending a known amount of B-caryophyllene through an ozone-containing reactor at different reaction times which makes the step pattern? Or is the reaction time in the flow tube fixed and you are changing the ozone amount. If the latter, can you show the ozone on the right y axis of this figure?

*As mentioned in the figure caption in more detail now, the reaction time stays constant, while the ozone concentration is varied. Ozone concentration was added to the plot.*

**Technical corrections**

Line 33: "..growth is critical.."✔

Line 43: need to cite Millet et al., 2018 correctly here and for subsequent citations to this article ✔

Figure 2: indicate the blue part is the blower ✔

Line 168: I think the use of "cross-talk" is a little misleading since you're referring to the interference of adjacent peaks. I suggest replacing "cross-talk" with "interference". ✔

Line 269-270: This sentence should be rewritten. One suggestion for how it could start: "A study that may serve as a better comparison was performed in…" ✔

Line 278: suggestion: "…frequent calibrations across multiple studies" ✔

Line 343: remove "were" ✔

Line 404: "leaf level" ✔

Figure 5: Bottom panel should have y axes and the y axis scale needs to include the whole graph. Middle panel should be described in caption. ✔

Line 483: I do not think you need the phrase "reserves in" in the section "…with the large reserves in flux signal to noise ratio…" or I do not understand what it means. ✔

Figure A1: need to reformat the legend to be a more legible time and include what the lines mean. The fitted curve equations also should be presented more neatly as well as designate which curve they are the equation of. ✔

*Plot and caption were updated.*
Line 515: "losing" instead of "loosing" ✔

---

## Author Comment (AC2)

**Reply on RC #2**

**Review of "First Eddy Covariance Flux Measurements of Semi Volatile Organic Compounds with the PTR3-TOF-MS", Fischer et al., AMT (2021)**

This paper presents observations of concentrations and EC fluxes of several biogenic VOC above a boreal forest. Measurements are acquired with a new instrument that has much higher sensitivity than its predecessors. They also use a novel inlet design that minimizes contact of sample gas with instrument surfaces. Measurements from a 3-week period in the spring are analyzed to demonstrate instrument performance and highlight the potential of the new capabilities. The paper is generally well written and the number and style of figures is appropriate. Publication is recommended after consideration of the following, mostly minor, critiques.

**General Comments**

Sections 4.5 and 4.6 deal with data quality and uncertainties. Possibly it makes more sense to present these before the discussion of specific fluxes in Sect. 4.1-4.4.

*The sections were moved to the experimental chapter and reorganized, as also requested by reviewer #2.*

**Specific Comments**

L87: It is stated that the inlet mounting was meant to minimize flow distortion. And later on L145, it is stated that the "anemometer deviation" is < 2 cm/s. What does this mean, precisely, and how was it determined? Were any tests carried out to confirm that flow distortion does not influence fluxes (e.g., comparing heat and momentum spectra/cospectra with the inlet flow on/off)? Also, is there potential systematic bias associated with such large sensor separation?

*We estimated the flow distortion at the sonic anemometer assuming that the flow is drawn into our inlet homogeneously from all directions. The air velocity through the surface of a corresponding sphere was calculated. We mention this now in the new version of the manuscript. We did not do spectral comparison with the inlet blower on and off, but since we expect the distortion to be a constant offset in the vertical component of the wind data, it is removed automatically by the wind tilt correction.*

*The flux attenuation effect of a purely vertical sensor separation is discussed by Horst et al. (2009). Their figure 11 shows measured flux deviation versus z/z' with z being the anemometer height and z' the scalar sensor height. Our inlet is 15m above the zero-displacement height with the anemometer positioned 0.5m higher, resulting in z/z'=1.033. Attenuation is smaller in this arrangement than with the anemometer situated below the scalar sensor, and neglectable for our z/z' compared to other uncertainties discussed in the manuscript. We added a short reference to Horst et al. (2009).*

*Horst, T.W., Lenschow, D.H. Attenuation of Scalar Fluxes Measured with Spatially-displaced Sensors. Boundary-Layer Meteorol 130, 275–300 (2009). https://doi.org/10.1007/s10546-008-9348-0*

L119 – 121: This sentence might be more appropriate for the introduction.

*This section was extended and is now part of the introduction as proposed also by reviewer #1.*

Figure 2: An actual photograph of the inlet/sonic setup, if available, would be helpful.

*We added a picture showing the inlet tube and the sonic anemometer to figure 2.*

L274 – 278 and 295 – 303: This is stated in the introduction and does not need to be reiterated here. Deletion or shortening suggested.

*The sections were shortened and merged with the corresponding sections in the introduction as proposed also by reviewer #1.*

L372: Is this a lower limit for the Bcary emission rate, since it does not account for deposition of the oxidation products?

*The section was partly rewritten, since we initially oversimplified the estimation of the reaction time and neglected in canopy reactions producing oxidation products. In the revised version we reduced the discussion of reacted Bcary. We see variations in reacted Bcary to total sesquiterpene flux ratio following ambient temperature. We speculate about in-canopy deposition losses, which could be a possible explanation of our observations.*

L388 – 392 – all of this description is more appropriate for the figure caption. Discussion here should focus on what the figure means, not the visual elements.

*The text was changed to describe more results and less figure details.*

Figure 6: It would be helpful to see this type of plot for all the fluxes (and maybe diel average mixing ratio, too?).

*We added a figure showing the discussed compounds (except of the unemployable monoterpene). It is placed in the appendix, since no new information is presented by the alternative representation of figure 5.*

L423 – 426: The exact same sentences, or nearly so, appear earlier in the manuscript (295 – 303).

*The duplicate was shortened here and in the monoterpene section to avoid repetition, since most is already covered in the introduction.*

Sect. 4.5: It would also be instructive to see the power spectra, either in the main text or supplement.

*Spectral analysis of our mass spectrometer data shows white noise. We suspect its origin in the Poisson counting statistics of the detector for each sample and quantization noise due to the low count rates. Antialiasing due to the short residence time in the drift tube compared to the sampling rate could also be a potential source. In the plot of f\*S against frequency, this noise contributes a lot to the mid and high frequency range of the power spectra depending on count rates, but it is uncorrelated to the vertical wind component. Cospectra and the covariances used for flux calculation are not affected by this out of phase noise. Therefore, power spectra will not aid in assessing high frequency damping of the covariance and are not shown. Here we give an example averaged from May 3$^{rd}$ to May 8$^{th}$ showing vertical wind, virtual temperature, isoprene, sesquiterpene and BCARY-O3. The spectra were normalized for better comparability.*

[Figure]

Figure 8: Would it be possible to add the w' – temperature lag correlations for comparison? These lag peaks seem very wide.

*The requested w'T' lag correlation was added to the plot and it looks similarly wide. The only obvious difference is the missing lag time of the maximum correlation, which was to be expected since both w and T are measured by the same instrument.*

**Technical Comments**

L33: "critical" ✔

L51: "predicted that" ✔

Figure 1: I am somewhat confused by the cartoons. What is the "measurement container" box meant to show? And the "inlet box", which seems to appear opposite of the stated inlet orientation?

*The white boxes indicate the orientation of the measurement container and the attached inlet structure to show why the red wind sector had to be excluded from analysis. The figure caption was clarified.*

L102: "empirical" ✔

L105: "expected" ✔

L149: What is the brand or manufacturer of the blower?

*The manufacturer of the blower is unknown, it was scavenged from an older setup without its housing.*

L351: it seems like pmol would be a more convenient unit here. ✔

Figure 4: why are concentrations shown in cps instead of pptv or similar mixing ratio units?

*The histograms were re-calculated with calibrated time traces and show pptv now.*

L421: "of the number of averaged samples" ✔

---

## Author Response (AR2)

**Associated Editor decision:**

**Publish subject to minor revisions (review by editor)**

**Comments to the author:**

The power spectrum figure provided in response to the second referee should be included as supplementary material and referenced in the main text. While it may or may not be true that this does not explain high-frequency damping as noted in the author response, it is still important to show this to demonstrate instrument performance. Such noise can lead to systematic bias and reduced precision in flux measurements - indeed, it is somewhat remarkable that the SQT_O3 cospectrum looks as good as it does.

*We added a chapter entitled "power spectra" in Appendix B containing figure B2 and a short description. The power spectra are mentioned in chapter 3.4 "Spectral analysis" referring to Appendix B.*